# Gaussian Belief Propagation Network for Depth Completion

## Abstract

Depth completion aims to predict a dense depth map from a color image with sparse depth measurements. Although deep learning methods have achieved state-of-the-art (SOTA), effectively handling the sparse and irregular nature of input depth data in deep networks remains a significant challenge, often limiting performance, especially under high sparsity. To overcome this limitation, we introduce the Gaussian Belief Propagation Network (GBPN), a novel hybrid framework synergistically integrating deep learning with probabilistic graphical models for end-to-end depth completion. Specifically, a scene-specific Markov Random Field (MRF) is dynamically constructed by the Graphical Model Construction Network (GMCN), and then inferred via Gaussian Belief Propagation (GBP) to yield the dense depth distribution. Crucially, the GMCN learns to construct not only the data-dependent potentials of MRF but also its structure by predicting adaptive non-local edges, enabling the capture of complex, long-range spatial dependencies. Furthermore, we enhance GBP with a serial & parallel message passing scheme, designed for effective information propagation, particularly from sparse measurements. Extensive experiments demonstrate that GBPN achieves SOTA performance on the NYUv2 and KITTI benchmarks. Evaluations across varying sparsity levels, sparsity patterns, and datasets highlight GBPN's superior performance, notable robustness, and generalizable capability.

## 1 Introduction

Dense depth estimation is critical for various computer vision and robotics tasks. While dedicated sensors or methods provide sparse, irregular depth measurements (Schonberger & Frahm, 2016), acquiring dense depth directly is often challenging or costly. Depth completion (DC) bridges this gap through inferring a dense depth map from sparse depth guided by a synchronized color image. Sparse depth provides essential scale and absolute constraints, while the high-resolution color image offers rich structural and semantic information necessary for propagating depth and filling missing regions. Consequently, DC is a vital technique attracting increasing interest for enhancing downstream applications like 3D object detection (Wu et al., 2022), novel view synthesis (Roessle et al., 2022), and robotic manipulation (Li et al., 2023).

Traditional DC methods mostly relied on hand-crafted pipelines (Ku et al., 2018) or fixed graphical models like Markov Random Fields (MRF) (Diebel & Thrun, 2005). Although offering some robustness, their rigid, pre-defined priors struggled to capture complex geometry and fine details. Recently, learning-based methods, primarily deep neural networks, have achieved impressive accuracy improvements by directly regressing dense depth from extracted features (Ma & Karaman, 2018). However, effectively processing sparse, irregular input within standard deep architectures remains a significant challenge (Uhrig et al., 2017; Huang et al., 2019; Tang et al., 2024), often degrading performance and robustness, particularly under high sparsity.

In this paper, we introduce the Gaussian Belief Propagation Network (GBPN), a novel hybrid framework that synergistically combines deep learning's representational power with the structured inference of probabilistic graphical models. Unlike methods that directly regress depth, GBPN trains a deep network (the Graphical Model Construction Network, GMCN) to dynamically construct a scene-specific MRF over dense depth variables. Dense depth is then efficiently inferred via Gaussian Belief Propagation (GBP) on the learned MRF.

This formulation offers several key advantages. Firstly, learning to construct a scene-specific MRF overcomes the limitations of fixed, hand-crafted models, enabling adaptation to diverse geometries. Secondly, sparse depth measurements are naturally integrated as principled data terms within the globally consistent MRF framework, inherently addressing input sparsity and irregularity by propagating depth across the entire image. Finally, GBPN yields a depth distribution, providing valuable confidence estimates for risk-aware downstream tasks, such as planning (Burns & Brock, 2007).

To realize this, our GMCN infers not only potentials of the MRF but also its structure by predicting non-local edges, allowing the model to adaptively capture complex, long-range spatial dependencies guided by image content. We also propose a novel serial & parallel message passing scheme for GBP to enhance information flow, particularly from sparse measurements to distant unmeasured pixels. The entire GBPN is trained end-to-end using a probability-based loss function leveraging the estimated mean and precision from GBP, promoting the learning of reliable depth predictions along with an estimate of their confidence.

We validate the efficacy of GBPN through extensive experiments on two leading DC benchmarks: NYUv2 for indoor scenes and KITTI for outdoor scenes. Our method achieves state-of-the-art performance on these datasets at the time of submission. Comprehensive ablation studies demonstrate the effectiveness of each component within GBPN, including the dynamic MRF construction, propagation scheme, and probability-based loss function, *etc*. Furthermore, evaluations across varying sparsity levels, sparsity patterns, and datasets highlight GBPN's superior performance, notable robustness, and generalizable capability. Code and trained models will be available at *omitted for blind review*.

## 2 RELATED WORK

Research on depth completion (DC) has undergone a significant evolution, transitioning from traditional hand-crafted techniques to data-driven learning methods. Notably, concepts and principles from traditional approaches have significantly influenced the design of recent learning-based methods. This section briefly reviews these two lines of work and the combination of their respective strengths.

**Traditional Hand-crafted Techniques.** Early DC approaches largely relied on explicitly defined priors and assumptions about scene geometry and texture, often borrowing techniques from traditional image processing and inpainting. These methods constructed hand-crafted pipelines to infer dense depth from sparse measurements. For example, Kopf et al. (2007) proposed joint bilateral filtering guided by color information to upsample low-resolution depth maps. Ku et al. (2018) employed a sequence of classical image processing operators, such as dilation and hole filling, for depth map densification. Zhao et al. (2021b) leveraged local surface smoothness assumptions to estimate depth by computing surface normals in a spherical coordinate system. Hawe et al. (2011) reconstructed dense disparity from sparse measurements by minimizing an energy function formulated by Compressive Sensing. Diebel & Thrun (2005) modeled depth completion as a multi-resolution Markov Random Field (MRF) with simple smoothness potentials, and then solved using conjugate gradient optimization. Chen & Koltun (2014) developed a global optimization approach to reconstruct dense depth modeled by MRF. While demonstrating effectiveness in certain scenarios, these hand-crafted methods often struggle to capture complex geometric structures and fine-grained details.

**Data-driven Learning Methods.** In contrast, learning-based methods typically formulate DC as an end-to-end regression task, taking color images and sparse depth maps as input and directly predicting the target dense depth map. A primary focus in this line of work has been on effectively handling the sparse input data and fusing information from different modalities (color and depth). For explicitly processing sparse data, pioneering work by Uhrig et al. (2017) introduced sparsity-invariant convolutions, designed to handle missing data by maintaining validity masks. Huang et al. (2019) integrated the sparsity-invariant convolutions into an encoder-decoder architecture. Eldesokey et al. (2019) extended this concept by propagating a continuous confidence measure. However, general-purpose network architectures employing standard convolutional layers, often combined with sophisticated multi-modal fusion strategies, have frequently demonstrated strong performance. For instance, Tang et al. (2020) proposed to learn content-dependent and spatially-variant kernels to guide the fusion of color and depth features. Zhang et al. (2023) leveraged Transformer architectures to capture long-range dependencies for feature extraction in depth completion. Chen et al. (2019) used 2D convolutions for image features and continuous convolutions for 3D point cloud features, followed by fusion. While achieving impressive results and learning complex mappings from data,

purely data-driven methods often exhibit limited generalization performance, particularly when faced with data of different sparsity levels.

**Combination of Models and Learning.** More recently, a significant trend has emerged towards approaches that combine the strengths of both traditional modeling and data-driven learning. These methods aim to leverage the benefits of learned features and powerful network architectures while incorporating explicit priors or structured inference mechanisms. A famous pioneering work is Liu et al. (2015), combining deep network with closed-form solver for monocular depth estimation. For depth completion, a prominent line of work, exemplified by CSPN-based methods (Cheng et al., 2019; 2020; Lin et al., 2022; Park et al., 2020), employs deep networks to predict parameters for a learned anisotropic diffusion process, refining the regressed depth from deep network. Wang et al. (2023a) integrated traditional image processing techniques to generate an initial depth map before applying a learned refinement module. Tang et al. (2024) learned a bilateral filter-like propagation process to effectively spread information from sparse measurements, Qu et al. (2020) combined deep learning with a least-squares solver to estimate depth with constraints derived from sparse measurements. Zuo & Deng (2024) formulated depth completion as a learned optimization problem, where a network predicts local depth differences used in an energy function iteratively minimized via conjugate gradient. However, these methods still struggle with sparse data processing and limited propagation range. Our method falls within this hybrid category, by learning a MRF and inferring it via GBP, inherently addressing the issues of input sparsity and irregularity.

## 3 THE PROPOSED METHOD

### 3.1 OVERVIEW

Given a color image $I \in \mathbb{R}^{H \times W \times 3}$, regardless of how sparse depth is measured—whether via active sensor, like LiDAR (Geiger et al., 2012), or passive method, like SFM (Schonberger & Frahm, 2016), or even interactive user guidance (Ron et al., 2018)—we can project these depth measurements onto the image plane to yield a sparse depth map $S \in \mathbb{R}^{H \times W}$, with the same resolution $(H, W)$ as $I$. The valid pixels in $S$ are typically irregularly distributed, and may vary significantly in number and location. Unlike most learning-based approaches that employ dedicated neural network layers to process $S$ (Eldesokey et al., 2019; Huang et al., 2019) and directly regress the dense depth map $X \in \mathbb{R}^{H \times W}$, as illustrated in Fig. 1, we formulate the dense depth estimation task as inference in a Markov Random Field (MRF) (in Section 3.2), and infer $X$ via Gaussian Belief Propagation (GBP) (in Section 3.3). In this formulation, $S$ serves as data term within the global optimization framework of MRF, which eliminates the need for designing specific neural network architectures tailored to processing sparse input data (Tang et al., 2024). Specifically, the MRF structure, particularly its edges, is dynamically generated by a graphical model construction network (in Section 3.4), depending on the input color image $I$ and optionally on intermediate estimates of the dense depth distribution. The entire framework is end-to-end trained with a probability-based loss function (in Section 3.5).

### 3.2 PROBLEM FORMULATION

We formulate the dense depth estimation problem using a Markov Random Field (MRF), *i.e.* a type of undirected graphical model $\mathcal{G} = (\mathcal{V}, \mathcal{E})$, where $\mathcal{V}$ is the set of nodes representing the image pixels and $\mathcal{E}$ is the set of pairwise edges connecting neighboring pixels. As shown in Fig. 1, each node $i \in \mathcal{V}$ is associated with a random variable $x_i$, representing the depth at pixel $i$. The joint probability distribution over the depth variables is defined according to the MRF structure:

$$p(X|I, S) \propto \prod_{i \in \mathcal{V}_v} \phi_i \prod_{(i,j) \in \mathcal{E}} \psi_{ij}, \tag{1}$$

where $\phi_i$ and $\psi_{ij}$ are the abbreviations for $\phi_i(x_i)$ and $\psi_{ij}(x_i, x_j)$, indicating unary and pairwise potentials respectively, and $\mathcal{V}_v$ is the set of nodes corresponding to pixels with a valid depth measurement in the sparse map $S$.

The *unary potential* $\phi_i$ defined for pixel $i \in \mathcal{V}_v$, constrains the quadratic distance between estimated depth $x_i$ and the depth measurement $s_i$, written as

$$\phi_i = \exp(-\frac{w_i(x_i - s_i)^2}{2}). \tag{2}$$

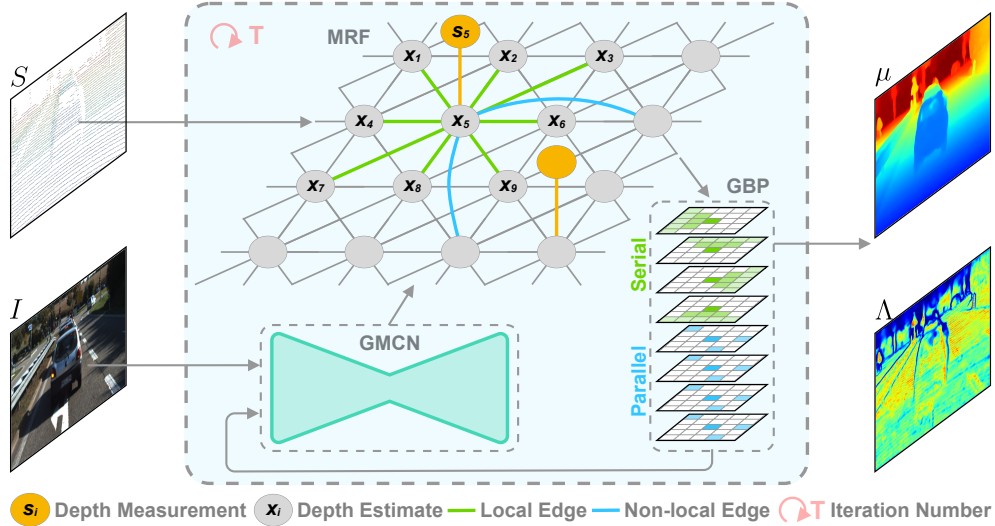

Figure 1: **Overview of the proposed approach.** Markov Random Field (MRF) is constructed depending on parameters dynamically generated from the Graphical Model Construction Network (GMCN), and then optimized via Gaussian Belief Propagation (GBP) for the distribution of dense depth map.

Here, $w_i > 0$ is a weight indicating the confidence for the depth measurement, and $\phi_i$ is only available if depth measurement $s_i$ is valid.

The *pairwise potential* $\psi_{ij}$ defined for edge $(i, j) \in \mathcal{E}$, encourages spatial coherence between the estimated depths of neighboring pixels $i$ and $j$, expressed as

$$\psi_{ij} = \exp(-\frac{w_{ij}(x_i - x_j - r_{ij})^2}{2}). \tag{3}$$

Here, $w_{ij} > 0$ is a weight controlling the strength of the spatial constraint, and $r_{ij}$ is the expected depth difference between $x_i$ and $x_j$ .

In traditional MRF-based approaches, the parameters $w$, and $r$ are typically hand-crafted based on simple assumptions or features. For instance, $r_{ij}$ is often set to $0$ to enforce simple smoothness (Diebel & Thrun, 2005), and $w_{ij}$ is commonly derived from local image features like color differences. These approaches, while providing some robustness, often limited the model's ability to capture complex scene geometry and fine details, leading to suboptimal results. In contrast, we employ an end-to-end trained deep network to dynamically construct the MRF based on the color image and optional optimized intermediate dense depth map distribution. This allows the MRF to adaptively model the scene, via placing stronger data constraints where measurements seem reliable, and applying sophisticated smoothness constraints that vary based on image content. Moreover, we further enhance the MRF's expressive power with *dynamic parameters* and *dynamic edges*.

**MRF with Dynamic Parameters.** The constructed MRF is approximately inferred in an iterative manner. Unlike traditional methods that define the MRF parameters a priori and keep them fixed during optimization, our approach updates the MRF parameters dynamically as the iterative inference progresses. This adaptive parameterization allows the MRF to evolve alongside the solution, making it more responsive to the current state of the variables and potentially improving both the solution quality and the convergence behavior of the optimization.

**MRF with Dynamic Edges.** Traditionally, the graph structure of an MRF, specifically the edges connecting a variable to its neighbors, is fixed and predefined, *e.g.* 8-connected local pixels for each variable. As illustrated in Fig. 1, alongside these fixed local edges (depicted in green), our framework also dynamically estimates and establishes non-local connections for each pixel (shown in blue). This dynamic graph structure allows the MRF to capture dependencies beyond immediate spatial neighbors, increasing its flexibility and capacity to model complex scene structures. Consequently, during inference, each variable's state is conditioned on both its predefined local neighbors and its dynamically generated non-local connections.

## 3.3 Approximate Inference Technology

### 3.3.1 Gaussian Belief Propagation

For a high-resolution image, computing the exact posterior for the above MRF is computationally prohibitive, due to the large number of variables and complex dependencies. Belief Propagation (BP) (Pearl, 1988) is a widely used algorithm for performing approximate probabilistic inference in graphical models. By iteratively operating on belief updating and message passing, BP enables parallel computation and often exhibits reasonably fast empirical convergence compared to other iterative inference methods (Weiss & Freeman, 1999). Though lacking formal convergence guarantees for general graphs, it has been successfully applied to various graphs, even with cycles (Freeman et al., 2000; Sun et al., 2003), known as loopy belief propagation. As introduced below and Section 3.3.2, we adopts damping tricks and graph decomposition to improve the stability and convergence.

In BP, the belief on variable $x_i$ is proportional to the product of its local evidence $\phi_i$ and incoming messages from all neighboring variables, written as

$$b_i \propto \phi_i \prod_{(i,j)\in\mathcal{E}} m_{j\to i}. \tag{4}$$

Messages are computed by marginalizing over the variable of the sending node, considering its local evidence and incoming messages from its other neighbors, expressed as

$$m_{j\to i} \propto \int_{x_j} \psi_{ij}\phi_j \prod_{(k,j)\in\mathcal{E}\setminus(i,j)} m_{k\to j}\, dx_j. \tag{5}$$

The formulated MRF in Section 3.2 is a Gaussian graph model, for which we can use Gaussian Belief Propagation (GBP) for inference. GBP (Bickson, 2008) assumes all beliefs $b$ and messages $m$ are under Gaussian distribution. This assumption allows the complex integration and product operations in eq. (5) to be simplified into algebraic updates on the parameters of the Gaussian distributions. We can take moment form $\mathcal{N}(\mu, \Lambda^{-1})$ or canonical form $\mathcal{N}^{-1}(\eta, \Lambda)$ as Gaussian parametrization for convenience, where $\eta = \mu\Lambda$. Then, in GBP, the belief updating corresponding to eq. (4) is given by:

$$\eta_i = w_i s_i + \sum_{(i,j)\in\mathcal{E}} \hat{\eta}_{j\to i},$$
$$\Lambda_i = w_i + \sum_{(i,j)\in\mathcal{E}} \hat{\Lambda}_{j\to i}. \tag{6}$$

And message passing corresponding to eq. (5) is given by:

$$\mu_{j\to i} = \mu_{j\setminus i} + r_{ij},$$
$$\Lambda_{j\to i}^{-1} = \Lambda_{j\setminus i}^{-1} + w_{ij}^{-1}. \tag{7}$$

Here, $j \setminus i$ means the set of all messages sent to $j$ except the message from $i$. We leave the detailed derivation of Gaussian belief propagation in the Section A.1.1. To improve convergence and stability, we adopt the *damping* trick for message passing (Murphy et al., 2013) by applying weighted average of the message from the previous iteration and the message computed in the current iteration, *i.e.* $\hat{m}_{j\to i,t} = \hat{m}_{j\to i,t-1}^{\beta_i} m_{j\to i,t}^{1-\beta_i}$. Once again corresponding to Gaussian parameters update, given by:

$$\hat{\eta}_{j\to i,t} = \beta_i \hat{\eta}_{j\to i,t-1} + (1-\beta_i)\eta_{j\to i,t},$$
$$\hat{\Lambda}_{j\to i,t} = \beta_i \hat{\Lambda}_{j\to i,t-1} + (1-\beta_i)\Lambda_{j\to i,t}. \tag{8}$$

### 3.3.2 Propagation Scheme

A crucial aspect of Belief Propagation is the definition of its message propagation scheme (Davison & Ortiz, 2019). Both serial and parallel propagation schemes offer distinct advantages. Serial propagation is effective at propagating information across the entire graph structure, allowing local evidence to influence distant nodes. This is particularly important for depth completion, ensuring that information from depth measurements propagates broadly, resulting in a dense depth estimation where every variable receives sufficient incoming messages to form a valid belief. In contrast, parallel

propagation, propagating messages over short ranges, is significantly more efficient by effectively leveraging modern hardware parallelism.

To combine both advantages, we design a hybrid serial & parallel propagation scheme. As illustrated in Fig. 1, our scheme splits the message passing into updates performed serially on directional local connections and updates performed in parallel on non-local connections, decomposing the loopy graph into loop-free sub-graphs. Specifically, we categorize edges $\mathcal{E}$ in MRF into four sets corresponding to local directional sweeps: left-to-right (LR), top-to-bottom (TB), right-to-left (RL), and bottom-to-top (BT), denoted $\mathcal{E}_{LR}, \mathcal{E}_{TB}, \mathcal{E}_{RL}$, and $\mathcal{E}_{BT}$ respectively, and a set for non-local connections, $\mathcal{E}_{NL}$. With this decomposition, each set of edges is with directionality and loop-free, which is more stable for convergence.

For serial propagation, we sequentially perform message passing sweeps utilizing these four local directional edge sets. Under each sweep direction, message and belief updates are processed serially according to a defined order (e.g., column by column for horizontal sweeps). For instance, during the sweep using edges $\mathcal{E}_{LR}$ (left-to-right), message updates for variables in column $n$ are computed only after updates related to column $n - 1$ have been completed. And the update for a variable at pixel $(m, n)$ incorporates messages received from specific neighbors in column $n - 1$, such as those connected via edges from $\mathcal{E}_{LR}$ linking $(m - 1, n - 1)$, $(m, n - 1)$, and $(m + 1, n - 1)$ to $(m, n)$. For non-local propagation, message updates for all pixels based on connections in $\mathcal{E}_{NL}$ are computed simultaneously. These non-local connections are dynamically generated to link pixels that are relevant but not necessarily spatially adjacent.

The whole propagation scheme is detailed in Algorithm 1. We initialize the $\eta$ and $\Lambda$ parameters for all messages on all relevant edges to $0$. We empirically set a fixed total number of iterations $T$. In each iteration, we first sequentially perform the four directional local serial propagation sweeps (LR, TB, RL, BT). Following the serial sweeps, we execute $T_n$ steps of parallel non-local propagation. Finally, after $T$ iterations, we yield the marginal beliefs for each pixel $i$. The estimated depth is the mean $\mu_i = \eta_i / \Lambda_i$ with the precision $\Lambda_i$ serving as a measure of confidence. The visualization of propagation and learned edges is introduced in Section A.3.

---

**Algorithm 1** Serial & Parallel Propagation Scheme

---

**Require:** Graph $\mathcal{G} = (\mathcal{V}, \mathcal{E})$. Edge subsets $\mathcal{E}_{LR}, \mathcal{E}_{TB}, \mathcal{E}_{RL}, \mathcal{E}_{BT}, \mathcal{E}_{NL}$.
**Ensure:** Estimated marginal belief represented by $\eta, \Lambda$.
1: **Initialization:**
2: **for all** edges $(i, j) \in \mathcal{E}$ **do**                               ▷ Initialize messages
3:     $\eta_{j \to i} = 0$
4:     $\Lambda_{j \to i} = 0$
5: **end for**
6: **Iterations:**
7: **for** $t = 1$ to $T$ **do**
8:     **for all** $\mathcal{E}_{LR}, \mathcal{E}_{TB}, \mathcal{E}_{RL}, \mathcal{E}_{BT}$ **do**                     ▷ Serial Propagation
9:         Serial message passing(eqs. (7) and (8)) and belief updating(eq. (6))
10:     **end for**
11:     **for** $t_n = 1$ to $T_n$ **do**                               ▷ Parallel Propagation
12:         message passing (eqs. (7) and (8)) for all $(i, j) \in \mathcal{E}_{NL}$ in parallel
13:         belief updating (eq. (6))
14:     **end for**
15: **end for**
16: **return** $\eta, \Lambda$

---

## 3.4 GRAPHICAL MODEL CONSTRUCTION NETWORK

As previously mentioned, the Markov Random Field (MRF) is dynamically constructed and subsequently optimized using Gaussian Belief Propagation (GBP). Our Graphical Model Construction Network is designed to learn the parameters and structures of this MRF from input data.

We employ a U-Net architecture (Ronneberger et al., 2015), comprising encoder and decoder layers, to extract multi-scale features essential for MRF construction. Within the U-Net, we introduce a novel

global-local processing unit. Each unit combines a dilated neighborhood attention layer (Hassani & Shi, 2022) and a ResNet block (He et al., 2016). The dilated neighborhood attention layer is utilized to capture long-range dependencies, effectively expanding the receptive field without increasing computational cost. As a complement, the ResNet block focuses on extracting and refining local features. More details about the network architecture is introduced in Section A.1.2.

We apply convolutional layers on the aggregated features to estimate the MRF parameters, such as $r$ and $w$. Additionally, the network estimates the damping rate $\beta$ used in the GBP and the offsets for constructing non-local neighboring pixels. Inspired by Deformable Convolutional Networks (Dai et al., 2017), we employ bilinear interpolation to sample Gaussian parameters at these non-local neighbor locations defined by the estimated float offsets. This approach ensures that gradients can be effectively backpropagated during the training phase. Considering that GBP typically converges to an accurate mean $\mu$ but not the exact precision $\Lambda$ (Weiss & Freeman, 1999), we also utilize convolution layers to estimate a residual term. This residual is added to the estimated $\Lambda$ from GBP, and a sigmoid activation function is then applied to yield the updated precision.

We provide two approaches to construct the MRF: a color image-only approach (termed GBPN-1) and a multi-modal fusion approach (termed GBPN-2) incorporating depth information. In GBPN-1, the U-Net architecture receives a color image as its sole input to construct the MRF. The GBPN-2 utilizes a second U-Net that takes both the color image and the optimized depth distribution from the GBPN-1 (*i.e.* $\mu$ and $\Lambda$) as input. For GBPN-2, cross-attention is leveraged within the dilated attention layers, where the query is generated from multi-modal features, while the keys and values are derived from the color image features in the first U-Net.

## 3.5 PROBABILITY-BASED LOSS FUNCTION

We adopt the combination of $L_1$ and $L_2$ loss as the loss on depth. For a pixel $i$, the loss on depth is

$$L_i^X = \frac{\|\mu_i - x_i^g\|_2^2 + \alpha \|\mu_i - x_i^g\|_1}{max(\|\mu - x^g\|_1)}, \tag{9}$$

where $x_i^g$ is the ground-truth depth and $\alpha$ is the hyperparameter to balance the $L_1$ and $L_2$ loss. The combined $L_1$ and $L_2$ loss is normalized with the maximum $L_1$ loss across the image, which makes the convergence more stable. As the output of our framework is the distribution of dense depth map, which is in Gaussian and can be represented with $\mu_i$ and $\Lambda_i$ for each pixel, we adopt the probability-based loss (Kendall & Gal, 2017) as the final loss:

$$L = \frac{1}{|\mathcal{V}_g|} \sum_{i \in \mathcal{V}_g} \Lambda_i L_i^X - \log(\Lambda_i). \tag{10}$$

Here, $\mathcal{V}_g$ denotes the set of pixels with valid ground-truth depth. In this way, the precision $\Lambda$ can also be directly supervised, without the need for its ground-truth value.

## 4 EXPERIMENTS

We evaluate our proposed method, GBPN, on two leading depth completion benchmarks: NYUv2 (Silberman et al., 2012) for indoor scenes and KITTI (Geiger et al., 2012) for outdoor scenes. To demonstrate its effectiveness, we provide comprehensive comparisons against state-of-the-art (SOTA) methods in Section 4.1. We then perform thorough ablation studies in Section 4.2 to analyze the contribution of core components. We also assess the robustness of GBPN by varying input sparsity levels in Section 4.3, a critical factor for real-world deployment. Finally, cross-dataset evaluations in Section 4.4 verify the model's generalization capability. The appendix contains further analysis on the experimental setup (Section A.2), noise sensitivity (Section A.6), and runtime efficiency (Section A.7).

## 4.1 COMPARISON WITH STATE-OF-THE-ART METHODS

We evaluate GBPN (the GBPN-2) on the official test sets of the NYUv2 (Silberman et al., 2012) dataset and the KITTI Depth Completion (DC) (Geiger et al., 2012) dataset. Quantitative comparisons

between GBPN and other top-performing published methods are presented in Table 1. On the KITTI DC benchmark[1], our method achieves the best iRMSE among all submissions at the time of writing, and demonstrates highly competitive performance across other evaluation metrics. Specifically, our method ranks second under RMSE among all published papers, where DMD3C (Liang et al., 2025) obtains the lowest RMSE. It's worth noting that DMD3C utilizes exact BP-Net (Tang et al., 2024) but incorporates additional supervision derived from a foundation model during training. In contrast, our method is trained solely on the standard KITTI training set from scratch, similar to BP-Net, while surpassing BP-Net in all evaluation metrics. Training with more data or distilling from a widely trained foundation model to improve our method is an interesting direction left for future work. On the NYUv2 dataset, our method achieves the best RMSE. As the commonly used $\delta_{1.25}$ metric is nearing saturation (typically $\geq 99.6\%$), we provide results using the stricter $\delta_{1.02}$ and $\delta_{1.05}$ metrics in Table 1 to better highlight the superiority of GBPN compared to other methods with publicly available models on the NYUv2 dataset. More comparisons can be found in Section A.4.

Table 1: **Performance on KITTI and NYUv2 datasets.** For the KITTI dataset, results are evaluated by the KITTI testing server. For the NYUv2 dataset, authors report their results in their papers. The best result under each criterion is in **bold**. The second best is with underline.

| | KITTI | | | | NYUv2 | | | |
|---|---|---|---|---|---|---|---|---|
| | RMSE↓ (mm) | MAE↓ (mm) | iRMSE↓ (1/km) | iMAE↓ (1/km) | RMSE↓ (m) | REL↓ | $\delta_{1.02}$ ↑ (%) | $\delta_{1.05}$ ↑ (%) |
| S2D (Ma & Karaman, 2018) | 814.73 | 249.95 | 2.80 | 1.21 | 0.230 | 0.044 | – | – |
| DeepLiDAR (Qiu et al., 2019) | 758.38 | 226.50 | 2.56 | 1.15 | 0.115 | 0.022 | – | – |
| GuideNet (Tang et al., 2020) | 736.24 | 218.83 | 2.25 | 0.99 | 0.101 | 0.015 | 82.0 | 93.9 |
| NLSPN (Park et al., 2020) | 741.68 | 199.59 | 1.99 | 0.84 | 0.092 | 0.012 | 88.0 | 95.4 |
| ACMNet (Zhao et al., 2021a) | 744.91 | 206.09 | 2.08 | 0.90 | 0.105 | 0.015 | – | – |
| DySPN (Lin et al., 2022) | 709.12 | 192.71 | 1.88 | 0.82 | 0.090 | 0.012 | – | – |
| BEV@DC (Zhou et al., 2023) | 697.44 | 189.44 | 1.83 | 0.82 | 0.089 | 0.012 | – | – |
| CFormer (Zhang et al., 2023) | 708.87 | 203.45 | 2.01 | 0.88 | 0.090 | 0.012 | 87.5 | 95.3 |
| LRRU (Wang et al., 2023a) | 696.51 | 189.96 | 1.87 | **0.81** | 0.091 | 0.011 | – | – |
| TPVD (Yan et al., 2024) | 693.97 | 188.60 | 1.82 | **0.81** | 0.086 | **0.010** | – | – |
| ImprovingDC (Wang et al., 2024) | 686.46 | **187.95** | 1.83 | **0.81** | 0.091 | 0.011 | – | – |
| OGNI-DC (Zuo & Deng, 2024) | 708.38 | 193.20 | 1.86 | 0.83 | 0.087 | 0.011 | 88.3 | 95.6 |
| BP-Net (Tang et al., 2024) | 684.90 | 194.69 | 1.82 | 0.84 | 0.089 | 0.012 | 87.2 | 95.3 |
| DMD3C (Liang et al., 2025) | **678.12** | 194.46 | 1.82 | 0.85 | **0.085** | 0.011 | – | – |
| GBPN | 682.20 | 192.14 | **1.78** | 0.82 | **0.085** | 0.011 | **89.1** | **95.9** |

## 4.2 ABLATION STUDIES

We conduct ablation studies on the NYUv2 dataset to evaluate the contribution of core components in our GBPN. Starting from a simple optimization-based baseline $V_1$, we incrementally add components to arrive at $V_2$ to $V_9$. All models are trained with half amount epochs of the full training schedule, due to resource constraints. Quantitative results, including RMSE and $\delta_{1.02}$, are presented in Table 2.

Table 2: Ablation studies on NYUv2 dataset.

| | Basic Block | | Local Edges | | Dynamic MRF | | GBP Iters. | | Loss | Criteria | |
|---|---|---|---|---|---|---|---|---|---|---|---|
| | Conv. | Attn. | 4 | 8 | Param. | Edge | 3 | 5 | Probability | RMSE↓(mm) | $\delta_{1.02}$ ↑(%) |
| $V_1$ | ✓ | | | | | | | | | 342.70 | 21.58 |
| $V_2$ | ✓ | | | | | | | | ✓ | 340.84 | 25.23 |
| $V_3$ | ✓ | | ✓ | | | | ✓ | | ✓ | 108.29 | 83.71 |
| $V_4$ | | ✓ | ✓ | | | | ✓ | | ✓ | 109.54 | 83.27 |
| $V_5$ | ✓ | ✓ | ✓ | | | | ✓ | | ✓ | 107.01 | 84.01 |
| $V_6$ | ✓ | ✓ | ✓ | | ✓ | | ✓ | | ✓ | 103.92 | 84.69 |
| $V_7$ | ✓ | ✓ | ✓ | | ✓ | ✓ | ✓ | | ✓ | 101.42 | 85.19 |
| $V_8$ | ✓ | ✓ | | ✓ | ✓ | ✓ | ✓ | | ✓ | 100.95 | 85.20 |
| $V_9$ | ✓ | ✓ | | ✓ | ✓ | ✓ | | ✓ | ✓ | 100.69 | 85.27 |

The baseline model $V_1$ employs a convolutional U-Net to estimate an initial depth and confidence map from RGB image, and then optimizes the parameters of an affine transformation using constraints from sparse depth measurements. Similar to Conti et al. (2023), this process involves solving a weighted least squares problem based solely on depth observation constraints, without any pair-wise terms. The model is trained using only an L2 loss on depth, following Tang et al. (2020; 2024).

---

[1] http://www.cvlibs.net/datasets/kitti/eval_depth.php?benchmark

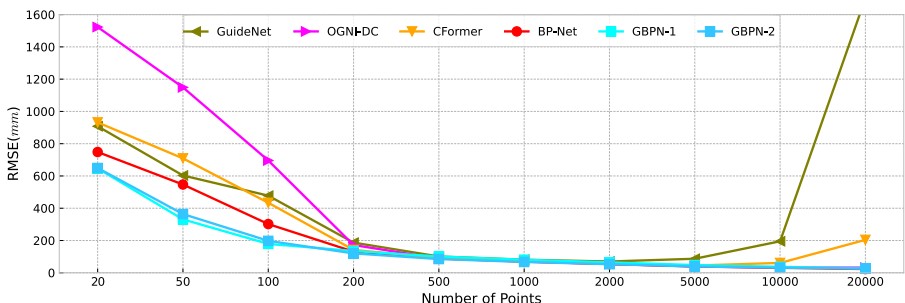

Figure 2: RMSE($mm$) on NYUv2 testset under various sparsity.

In our ablation study, we first replace the L2 loss in $V_1$ with our proposed optimization-based loss function, resulting in $V_2$ with slight performance improvement. Building on $V_2$, we develop $V_3$, $V_4$, and $V_5$, which generate depth by solving a fixed MRF with fixed local edges, but using features from different backbones. All three variants outperform $V_2$ by a large margin, with $V_5$ achieving the best RMSE. This confirms the effectiveness of our problem formulation and the proposed global-local processing unit, which combines local convolutions with long-range attention. We then extend $V_5$ by incorporating an MRF with dynamic parameters ($V_6$) and dynamic non-local edges ($V_7$). The performance gains of $V_6$ and $V_7$ demonstrate the advantage of a more expressive graph model. Subsequently, we increase the number of local edges in $V_7$ to construct $V_8$, and then the number of iterations to form $V_9$. The performance improvements seen in $V_8$ and $V_9$ highlight the benefits of more dense constraints and a greater number of optimization iterations. We attempt further increases iteration numbers but observing little performance improvement, and finally choose $V_9$ as GBPN-1.

## 4.3 Sparsity Robustness Analysis

In real-world applications, the sparsity level of the input depth map may vary significantly depending on the sensor and environment. To evaluate the robustness of our method on input sparsity, we conducte experiments on the NYUv2 dataset, comparing our approach against other SOTA methods with publicly available code and models. For this analysis, all methods for comparison are from released models trained with 500 depth points on NYUv2. These models are then directly evaluated on sparse inputs generated at various sparsity levels, ranging from 20 to 20,000 points.

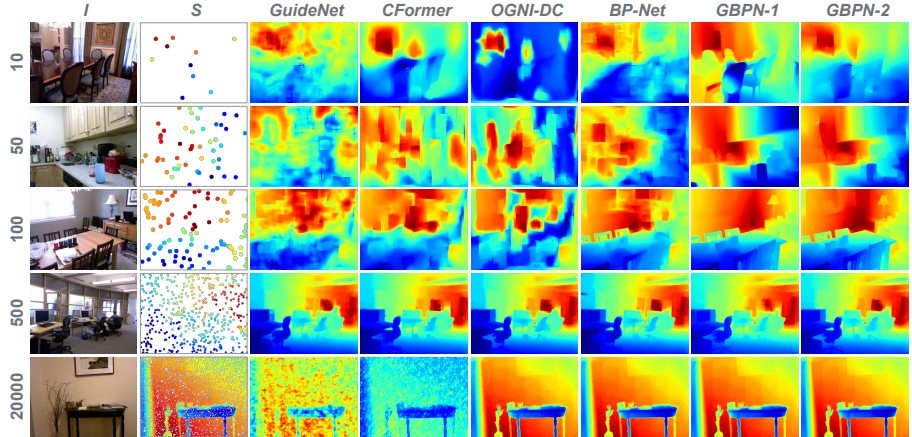

Figure 3: **Qualitative comparisons under different sparsity.** All comparing methods are trained with 500 depth points and directly tested with depth input from various sparsity levels.

As demonstrated in Fig. 2, both GBPN-1 and GBPN-2 exhibit consistently better robustness across the entire range of tested sparsity levels than others. For GBPN, the RMSE consistently decreases

as the number of points increases. In contrast, the RMSE of methods like GuideNet (Tang et al., 2020) and CFormer (Zhang et al., 2023) increases notably when the number of depth points becomes significantly denser (beyond approximately 5000 points) than the training sparsity (500 points). More about sparsity robustness analysis is in Section A.5.

We also visualize qualitative results from these compared methods under various sparsity levels in Fig. 3. In very sparse scenarios, such as the 10 points case visualized in the first row, depth maps from our method maintain relatively sharp boundaries, whereas the results from other methods appear ambiguous or messy. For the 50 and 100 points cases shown in the second and third rows, our method can produce relatively clear depth maps with structural details, while results from others tend towards being unclear and oversmooth. In very dense scenarios, *i.e.* 20,000 points case in the last row, our method consistently produces very clear depth maps, while results from some comparison methods tend to be noisy. We attribute this superior robustness to our problem formulation, which incorporates sparse input data as a data term within a globally optimized MRF framework, making it inherently robust to the irregularity and varying density of the input sparse measurements.

## 4.4 GENERALIZATION CAPABILITY

To demonstrate the generalization of GBPN, we evaluate state-of-the-art depth completion methods on the VOID (Wong et al., 2020) benchmark, whose sparse depth is from visual odometry. All methods are trained on NYUv2 with 500 random points and are evaluated zero-shot on the VOID validation set under 150, 500, and 1500 point settings. Thus, this setup tests generalization across different sparsity levels, sparsity patterns, and scene domains.

Table 3: Depth completion results on the VOID dataset at different sparsity levels (1500, 500, 150 points). RMSE and MAE are reported in meters.

| Method | VOID 1500 | | VOID 500 | | VOID 150 | |
|---|---|---|---|---|---|---|
| | RMSE (m) | MAE (m) | RMSE (m) | MAE (m) | RMSE (m) | MAE (m) |
| OGNI (Zuo & Deng, 2024) | 0.92 | 0.39 | 1.10 | 0.61 | 1.25 | 0.75 |
| NLSPN (Park et al., 2020) | 0.69 | **0.22** | 0.76 | **0.30** | 0.93 | 0.43 |
| CFormer (Zhang et al., 2023) | 0.73 | 0.26 | 0.82 | 0.38 | 0.96 | 0.48 |
| BP-Net (Tang et al., 2024) | 0.74 | 0.27 | 0.80 | 0.37 | 0.94 | 0.47 |
| GuideNet (Tang et al., 2020) | 2.14 | 1.05 | 2.06 | 0.85 | 2.05 | 0.70 |
| GBPN | **0.68** | **0.22** | **0.74** | **0.30** | **0.90** | **0.41** |

As reported in Table 3, GBPN consistently surpasses other methods across all sparsity levels in both RMSE and MAE. We believe this robustness stems from our model's formulation, which integrates sparse depth as principled observation terms within an MRF. By avoiding specialized neural layers for sparse input, our method achieves stronger generalization. In contrast, GuideNet (Tang et al., 2020) performs poorly on this dataset, perhaps due to its direct application of standard convolutional layers on sparse inputs.

## 5 CONCLUSION

This paper presents the Gaussian Belief Propagation Network (GBPN), a novel framework that seamlessly integrates deep learning with probabilistic graphical models for depth completion. GBPN employs a Graphical Model Construction Network (GMCN) to dynamically build a scene-specific Markov Random Field (MRF). This learned MRF formulation naturally incorporates sparse data and models complex spatial dependencies through adaptive non-local edges and dynamic parameters. Dense depth inference is efficiently performed using Gaussian Belief Propagation, enhanced by a serial & parallel message passing scheme. Extensive experiments on the NYUv2, KITTI, and VOID benchmarks demonstrate that GBPN achieves state-of-the-art accuracy and exhibits superior robustness and generalizable capability. Training on larger, more diverse datasets or leveraging supervision from pre-trained foundation models and optimize the GBP implementation are promising avenues for future works to enhance performance and efficiency for applications in broader scenarios.

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

## A  APPENDIX

### A.1  ADDITIONAL METHOD DETAILS

#### A.1.1  GAUSSIAN BELIEF PROPAGATION

This section provides a detailed derivation of the Gaussian Belief Propagation (GBP) equations, which are summarized in the main paper due to page limitations. Following the formulation in the

main paper, we model the joint probability distribution over the depth variables $X$ using a Markov Random Field (MRF):

$$p(X|I,S) \propto \prod_{i \in \mathcal{V}_v} \phi_i \prod_{(i,j) \in \mathcal{E}} \psi_{ij}, \tag{11}$$

where $\phi_i$ is the unary potential for node $i$, and $\psi_{ij}$ is the pairwise potential between node $i$ and $j$. Specifically, we define the unary potential as:

$$\phi_i = \exp(-\frac{w_i(x_i - s_i)^2}{2}), \tag{12}$$

and the pairwise potential as:

$$\psi_{ij} = \exp(-\frac{w_{ij}(x_i - x_j - r_{ij})^2}{2}). \tag{13}$$

In Belief Propagation, the belief at each node $i$ is calculated as the product of the local unary potential and all incoming messages from neighboring nodes

$$b_i \propto \phi_i \prod_{(i,j) \in \mathcal{E}} m_{j \to i}. \tag{14}$$

And the messages are updated iteratively according to

$$m_{j \to i} \propto \int_{x_j} \psi_{ij} \phi_j \prod_{(k,j) \in \mathcal{E} \setminus (i,j)} m_{k \to j} \, dx_j. \tag{15}$$

All the above eqs. (11) to (15) are same as eqs. (1) to (5) in the main paper. We repeat them here for reading convenience.

In GBP, all beliefs and messages are assumed to be Gaussian distributions. Then, a message proportional to a Gaussian, can be written as

$$m_{j \to i} \propto \exp(-\frac{\hat{\Lambda}_{j \to i}}{2} x_i^2 + \hat{\eta}_{j \to i} x_i). \tag{16}$$

Substituting the unary potential eq. (12) and the incoming messages eq. (16) into the belief equation eq. (14), we derive the belief

$$
\begin{aligned}
b_i &\propto \phi_i \prod_{(i,j) \in \mathcal{E}} m_{j \to i} \\
&\propto \exp(-\frac{w_i(x_i - s_i)^2}{2}) \prod_{(i,j) \in \mathcal{E}} \exp(-\frac{\hat{\Lambda}_{j \to i}}{2} x_i^2 + \hat{\eta}_{j \to i} x_i) \\
&\propto \exp(-\frac{1}{2}(w_i + \sum_{(i,j) \in \mathcal{E}} \hat{\Lambda}_{j \to i}) x_i^2 + (w_i s_i + \sum_{(i,j) \in \mathcal{E}} \hat{\eta}_{j \to i}) x_i).
\end{aligned} \tag{17}
$$

Thus, the resulting belief $b_i$ is also Gaussian, expressed as $\mathcal{N}^{-1}(\eta_i, \Lambda_i)$, with

$$
\begin{aligned}
\eta_i &= w_i s_i + \sum_{(i,j) \in \mathcal{E}} \hat{\eta}_{j \to i}, \\
\Lambda_i &= w_i + \sum_{(i,j) \in \mathcal{E}} \hat{\Lambda}_{j \to i}.
\end{aligned} \tag{18}
$$

The belief parameters in eq. (18) are identical to the eq. (6) of the main paper.

We can also define $b_{j \setminus i}$ as the belief on node $j$ taking all messages, except the message from $i$. Thus, similar to eq. (17), we have

$$b_{j \setminus i} \propto \phi_j \prod_{(k,j) \in \mathcal{E} \setminus (i,j)} m_{k \to j}. \tag{19}$$

And similar to eq. (18), $b_{j \setminus i}$ is in Gaussian $\mathcal{N}^{-1}(\eta_{j \setminus i}, \Lambda_{j \setminus i})$, with

$$\eta_{j \setminus i} = w_j s_j + \sum_{(k,j) \in \mathcal{E} \setminus (i,j)} \hat{\eta}_{k \to j},$$
$$\Lambda_{j \setminus i} = w_j + \sum_{(k,j) \in \mathcal{E} \setminus (i,j)} \hat{\Lambda}_{k \to j}. \tag{20}$$

Now, substituting the pairwise potential eq. (13) and the expression for $b_{j \setminus i}$ (from combining eq. (19) and eq. (20)) into the message update equation eq. (15), we get:

$$m_{j \to i} \propto \int_{x_j} \psi_{ij} \phi_j \prod_{(k,j) \in \mathcal{E} \setminus (i,j)} m_{k \to j} \, dx_j$$

$$\propto \int_{x_j} \psi_{ij} b_{j \setminus i} \, dx_j$$

$$\propto \int_{x_j} \exp(-\frac{w_{ij}(x_i - x_j - r_{ij})^2}{2}) \exp(-\frac{\Lambda_{j \setminus i}}{2} x_j^2 + \eta_{j \setminus i} x_j) \, dx_j$$

$$\propto \exp(-\frac{w_{ij}}{2} x_i^2 + w_{ij} r_{ij} x_i) \int_{x_j} \exp(-\frac{w_{ij}}{2} x_j^2 + (x_i - r_{ij}) w_{ij} x_j) \exp(-\frac{\Lambda_{j \setminus i}}{2} x_j^2 + \eta_{j \setminus i} x_j) \, dx_j$$

$$\propto \exp(-\frac{w_{ij}}{2} x_i^2 + w_{ij} r_{ij} x_i) \int_{x_j} \exp(-\frac{w_{ij} + \Lambda_{j \setminus i}}{2} x_j^2 + (x_i w_{ij} - r_{ij} w_{ij} + \eta_{j \setminus i}) x_j) \, dx_j \tag{21}$$

$$\propto \exp(-\frac{w_{ij}}{2} x_i^2 + w_{ij} r_{ij} x_i) \exp(\frac{(x_i w_{ij} - r_{ij} w_{ij} + \eta_{j \setminus i})^2}{2(w_{ij} + \Lambda_{j \setminus i})})$$

$$\propto \exp(-\frac{w_{ij}}{2} x_i^2 + w_{ij} r_{ij} x_i) \exp(\frac{(x_i w_{ij} - r_{ij} w_{ij} + \Lambda_{j \setminus i} \mu_{j \setminus i})^2}{2(w_{ij} + \Lambda_{j \setminus i})})$$

$$\propto \exp(-\frac{w_{ij} \Lambda_{j \setminus i}}{2(w_{ij} + \Lambda_{j \setminus i})} x_i^2 + \frac{w_{ij} \Lambda_{j \setminus i}}{w_{ij} + \Lambda_{j \setminus i}} (\mu_{j \setminus i} + r_{ij}) x_i)$$

$$\propto \exp(-\frac{(x_i - (\mu_{j \setminus i} + r_{ij}))^2}{2(w_{ij}^{-1} + \Lambda_{j \setminus i}^{-1})})$$

Finally, the message $m_{j \to i}$ is in Gaussian $\mathcal{N}(\mu_{j \to i}, \Lambda_{j \to i}^{-1})$, with

$$\mu_{j \to i} = \mu_{j \setminus i} + r_{ij},$$
$$\Lambda_{j \to i}^{-1} = \Lambda_{j \setminus i}^{-1} + w_{ij}^{-1}. \tag{22}$$

The eq. (22) here is identical to the eq. (7) of the main paper.

### A.1.2 GRAPHICAL MODEL CONSTRUCTION NETWORK

The Graphical Model Construction Network (GMCN) is designed to learn the parameters and structures of the MRF from input data. We employ a U-Net architecture (Ronneberger et al., 2015), comprising encoder and decoder layers, as the GMCN. The detailed network architecture of GMCN with $256 \times 320$ input images is provided in Table 4. Note that for clarity, only the main operators are listed here, with trivial operations like normalization, activation, and skip connections omitted. The encoder of the U-Net extracts features at six different scales ($\mathbf{E}^0, ..., \mathbf{E}^5$). From scale 1 to 5, feature extraction is performed using global-local units, which combine a dilated neighbor attention layer (Hassani & Shi, 2022) to capture long-range dependencies with ResBlocks (He et al., 2016) for robust local feature learning. The decoder then aggregates these multi-resolution features back to the original input resolution using deconvolution layers and concatenation operations. This multi-scale approach ensures that the network captures both fine-grained local details and broader contextual information necessary for robust MRF construction. We explore two variations of the GMCN. The first takes only color images as input, constructing the graphical model solely based on visual cues using a single U-Net. The second incorporates intermediate optimized dense depth distributions as

Table 4: Detailed Architecture of Graphical Model Construction Network (GMCN).

| | Output | Input | Operator | Output Size |
|---|---|---|---|---|
| | $\mathbf{E}_1^0$ | $I$ | Conv. + ResBlock | $(32, 256, 320)$ |
| | $\mathbf{E}_1^1$ | $\mathbf{E}_1^0$ | Conv. + (Self-Attn. + ResBlock) $\times 2$ | $(64, 128, 160)$ |
| | $\mathbf{E}_1^2$ | $\mathbf{E}_1^1$ | Conv. + (Self-Attn. + ResBlock) $\times 2$ | $(128, 64, 80)$ |
| | $\mathbf{E}_1^3$ | $\mathbf{E}_1^2$ | Conv. + (Self-Attn. + ResBlock) $\times 2$ | $(256, 32, 40)$ |
| Color | $\mathbf{E}_1^4$ | $\mathbf{E}_1^3$ | Conv. + (Self-Attn. + ResBlock) $\times 2$ | $(256, 16, 20)$ |
| Image | $\mathbf{E}_1^5$ | $\mathbf{E}_1^4$ | Conv. + (Self-Attn. + ResBlock) $\times 2$ | $(256, 8, 10)$ |
| Only | $\mathbf{D}_1^4$ | $\mathbf{E}_1^5, \mathbf{E}_1^4$ | Deconv. + Concat. + Conv. | $(256, 16, 20)$ |
| GMCN | $\mathbf{D}_1^3$ | $\mathbf{D}_1^4, \mathbf{E}_1^3$ | Deconv. + Concat. + Conv. | $(256, 32, 40)$ |
| | $\mathbf{D}_1^2$ | $\mathbf{D}_1^3, \mathbf{E}_1^2$ | Deconv. + Concat. + Conv. | $(256, 64, 80)$ |
| | $\mathbf{D}_1^1$ | $\mathbf{D}_1^2, \mathbf{E}_1^1$ | Deconv. + Concat. + Conv. | $(64, 128, 160)$ |
| | $\mathbf{D}_1^0$ | $\mathbf{D}_1^1, \mathbf{E}_1^0$ | Deconv. + Concat. + Conv. | $(32, 256, 320)$ |
| | $\beta_1, r_1, w_1, o_1$ | $\mathbf{D}_1^0$ | Conv. + Conv. | $(*, 256, 320)$ |
| GBP | $\mu_1, \Lambda_1$ | $S, \beta_1, r_1, w_1, o_1$ | Serial & Parallel Propagation | $(*, 256, 320)$ |
| | $\mathbf{E}_2^0$ | $I, \mu_1, \Lambda_1$ | Conv. + ResBlock | $(32, 256, 320)$ |
| | $\mathbf{E}_2^1$ | $\mathbf{E}_2^0, \mathbf{E}_1^1$ | Conv. + (Self/Cross-Attn. + ResBlock) $\times 4$ | $(64, 128, 160)$ |
| | $\mathbf{E}_2^2$ | $\mathbf{E}_2^1, \mathbf{E}_1^2$ | Conv. + (Self/Cross-Attn. + ResBlock) $\times 4$ | $(128, 64, 80)$ |
| | $\mathbf{E}_2^3$ | $\mathbf{E}_2^2, \mathbf{E}_1^3$ | Conv. + (Self/Cross-Attn. + ResBlock) $\times 4$ | $(256, 32, 40)$ |
| Multi | $\mathbf{E}_2^4$ | $\mathbf{E}_2^3, \mathbf{E}_1^4$ | Conv. + (Self/Cross-Attn. + ResBlock) $\times 4$ | $(256, 16, 20)$ |
| Modal | $\mathbf{E}_2^5$ | $\mathbf{E}_2^4, \mathbf{E}_1^5$ | Conv. + (Self/Cross-Attn. + ResBlock) $\times 4$ | $(256, 8, 10)$ |
| Fusion | $\mathbf{D}_2^4$ | $\mathbf{E}_2^5, \mathbf{E}_2^4$ | Deconv. + Concat. + Conv. | $(256, 16, 20)$ |
| GMCN | $\mathbf{D}_2^3$ | $\mathbf{D}_2^4, \mathbf{E}_2^3$ | Deconv. + Concat. + Conv. | $(256, 32, 40)$ |
| | $\mathbf{D}_2^2$ | $\mathbf{D}_2^3, \mathbf{E}_2^2$ | Deconv. + Concat. + Conv. | $(256, 64, 80)$ |
| | $\mathbf{D}_2^1$ | $\mathbf{D}_2^2, \mathbf{E}_2^1$ | Deconv. + Concat. + Conv. | $(64, 128, 160)$ |
| | $\mathbf{D}_2^0$ | $\mathbf{D}_2^1, \mathbf{E}_2^0$ | Deconv. + Concat. + Conv. | $(32, 256, 320)$ |
| | $\beta_2, r_2, w_2, o_2$ | $\mathbf{D}_2^0$ | Conv. + Conv. | $(*, 256, 320)$ |
| GBP | $\mu_2, \Lambda_2$ | $S, \beta_2, r_2, w_2, o_2$ | Serial & Parallel Propagation | $(*, 256, 320)$ |

additional input to refine the constructed graphical model. This is achieved by introducing a second multi-modal fusion U-Net, which utilizes a cross-attention mechanism to effectively integrate features from the color-based U-Net and the depth information.

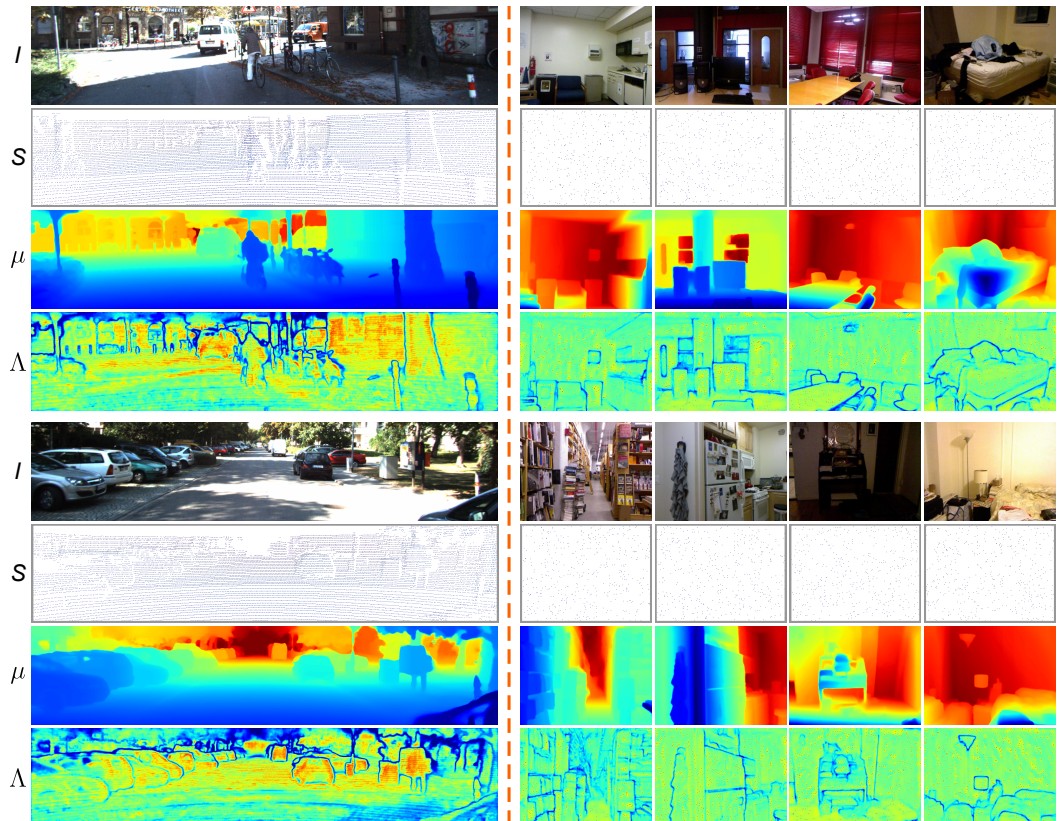

Figure 4: Visualization of estimated depth in Gaussian with $\mu$ and $\Lambda$. The results on KITTI are on the left, and results on NYUv2 are on the right.

## A.2 EXPERIMENTS SETUP

### A.2.1 DATASETS

We adopt the standard depth completion datasets, *i.e.* NYUv2 for indoor scenes and KITTI for outdoor scenes, to train and evaluate our method. We also evaluate our method on the VOID (Wong et al., 2020) dataset to validate the robustness and generalization.

The NYUv2 dataset (Silberman et al., 2012) comprises 464 scenes captured by a Kinect sensor. For training, we take the data proposed by Ma & Karaman (2018), utilizing 50,000 frames sampled from 249 scenes. Evaluation is performed on the official test set, which contains 654 samples from 215 distinct scenes. We follow common practice (Tang et al., 2020; Park et al., 2020; Zhang et al., 2023; Tang et al., 2024) to process data. Input images are initially down-sampled to $240 \times 320$ and then center-cropped to a resolution of $228 \times 304$. Sparse depth maps are generated for each frame by randomly sampling 500 points from the ground truth depth map.

The KITTI depth completion (DC) dataset (Uhrig et al., 2017) was collected using an autonomous driving platform. Ground truth depth is derived from temporally aggregated LiDAR scans and further refined using stereo image pairs. The dataset provides 86,898 frames for training and 1000 frames for validation. An additional 1000 test frames are evaluated on a remote server with a public leaderboard[2]. During training, we randomly crop frames to $256 \times 1216$. For testing, full-resolution frames are used.

The VOID dataset (Wong et al., 2020) was collected using an Intel RealSense D435i camera, with sparse measurements by a visual odometry system at 3 different sparsity levels, *i.e.*, 1500, 500, and 150 points, corresponding to $0.5\%$, $0.15\%$, and $0.05\%$ density. Each test split contains 800 images at $480 \times 640$ resolution. Though the dataset is collected for indoor scenes, the scenes, depth

---

[2]http://www.cvlibs.net/datasets/kitti/eval_depth.php?benchmark

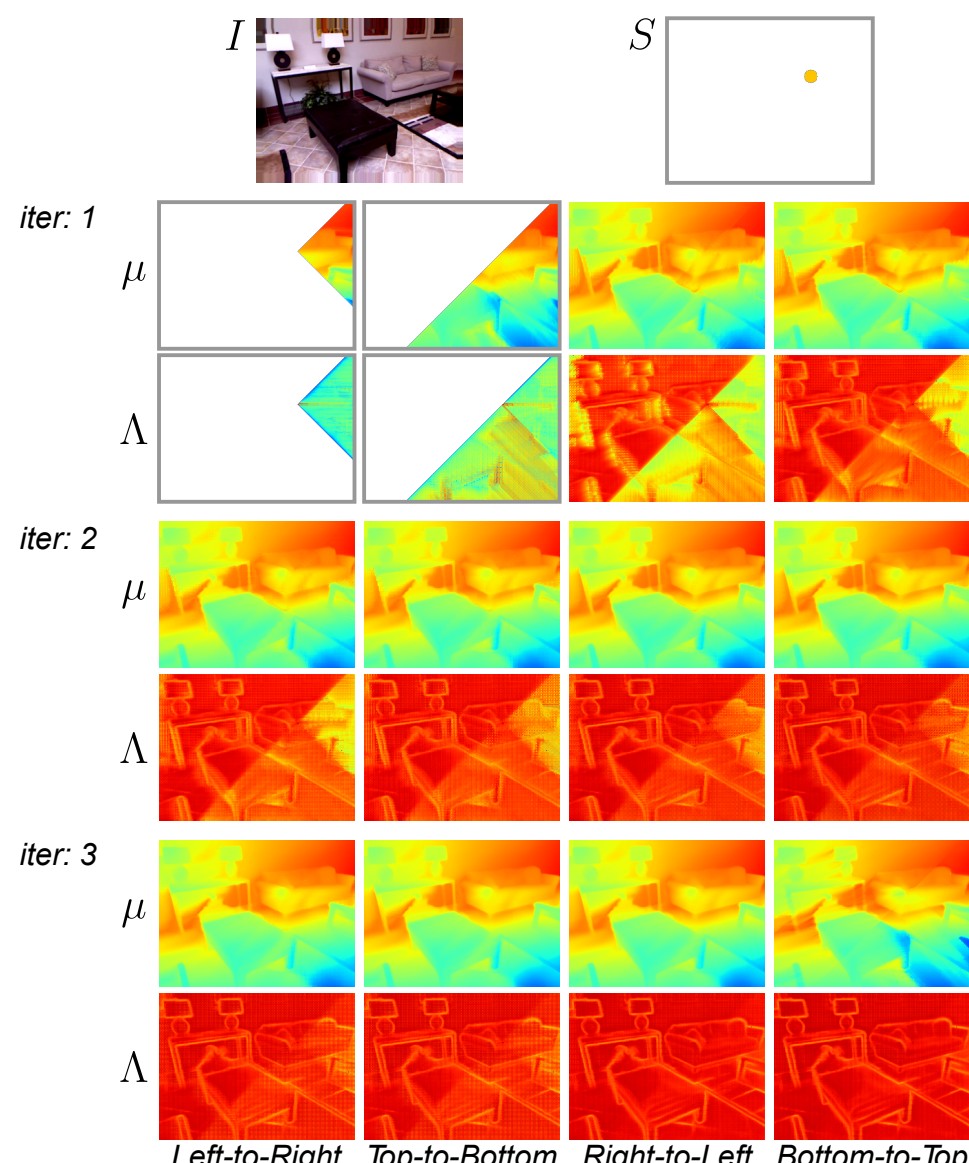

Figure 5: Visualization of intermediate results under the serial propagation of Gaussian Belief Propagation. Messages are propagated iteratively corresponding to local directional sweeps: left-to-right, top-to-bottom, right-to-left, and bottom-to-top.

measurement manner and depth sparsity in this dataset are distinct from NYUv2. As the collected ground truth data is noisy, this dataset is not used for training, but only for evaluation to validate the zero-shot generalization of DC methods to different sparsity patterns and scenes.

### A.2.2 TRAINING DETAILS

Our method is implemented in PyTorch and trained on a workstation with 4 NVIDIA RTX 4090 GPUs. We incorporate DropPath (Larsson et al., 2016) before residual connections as a regularization technique. We use the AdamW optimizer (Loshchilov & Hutter, 2018) with a weight decay of 0.05 and apply gradient clipping with an L2-norm threshold of 0.1. Models are trained from scratch for approximately 300,000 iterations. We employ the OneCycle learning rate policy (Smith & Topin, 2019), where the learning rate is annealed to 25% of its peak value during the cycle. For KITTI, the batch size is 8 and the peak learning rate is 0.001. For NYUv2 dataset, we use a batch size of 16

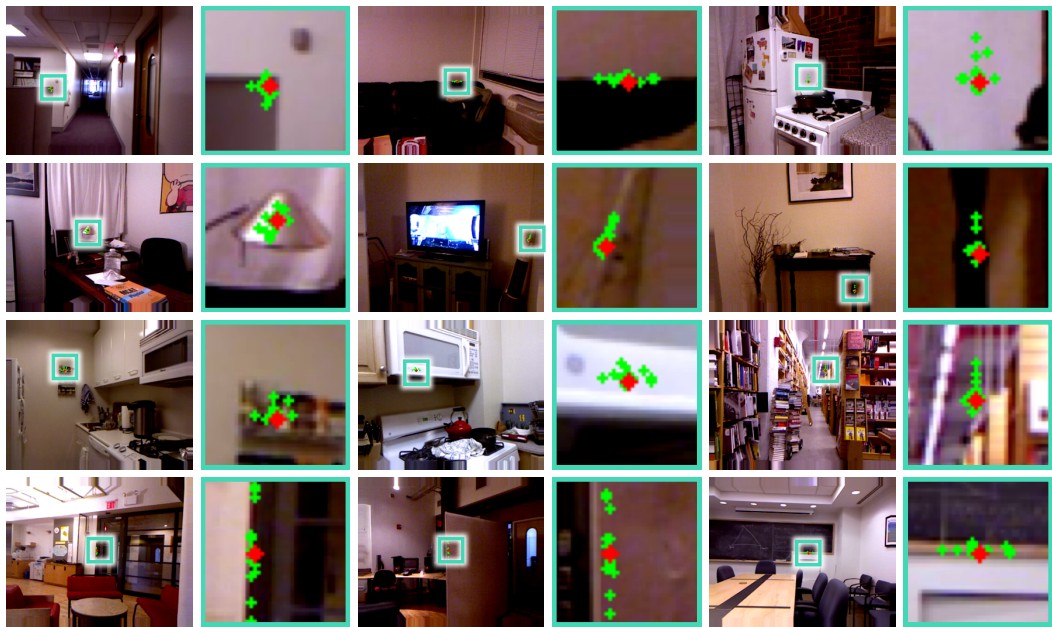

Figure 6: Visualization of dynamically constructed non-local edges in MRF. The red point is the target node, and the green points are the dynamically constructed neighbor nodes.

and a peak learning rate of $0.002$. The final model weights are obtained using Exponential Moving Average (EMA).

### A.2.3 DETAILS ON EVALUATION METRICS

We verify our method on both indoor and outdoor scenes with standard evaluation metrics. For indoor scenes, root mean squared error (RMSE), mean absolute relative error (REL), and $\delta_\theta$ are chosen as evaluation metrics. For outdoor scenes, the standard evaluation metrics are root mean squared error (RMSE), mean absolute error (MAE), root mean squared error of the inverse depth (iRMSE), and mean absolute error of the inverse depth (iMAE). These evaluation metrics are firstly calculated on each sample and then averaged among samples. And for each sample, they can be written as:

$$
\begin{aligned}
RMSE &= \left(\tfrac{1}{n}\sum_{i\in\mathcal{V}_g}(x_i^g - x_i)^2\right)^{\frac{1}{2}}, \\
iRMSE &= \left(\tfrac{1}{n}\sum_{i\in\mathcal{V}_g}\left(\tfrac{1}{x_i^g} - \tfrac{1}{x_i}\right)^2\right)^{\frac{1}{2}}, \\
MAE &= \tfrac{1}{n}\sum_{i\in\mathcal{V}_g}\left|x_i^g - x_i\right|, \\
iMAE &= \tfrac{1}{n}\sum_{i\in\mathcal{V}_g}\left|\tfrac{1}{x_i^{gt}} - \tfrac{1}{x_i}\right|, \\
REL &= \tfrac{1}{n}\sum_{i\in\mathcal{V}_g}\tfrac{|x_i^{gt}-x_i|}{x_i^g}, \\
\delta_\theta &= \tfrac{1}{n}\sum_{i\in\mathcal{V}_g}\left|\left\{max(\tfrac{x_i^g}{x_i},\tfrac{x_i}{x_i^g}) < \theta\right\}\right|.
\end{aligned}
\tag{23}
$$

Here, $\mathcal{V}_g$ is the set of pixels with valid ground truth, and $n = |\mathcal{V}_g|$ is the size of the set.

### A.3 VISUALIZATION

The output of our Gaussian Belief Propagation Network (GBPN) is a dense depth distribution, modeled as a Gaussian and represented by its mean $\mu$ and precision $\Lambda$. This precision $\Lambda$ can serve as a confidence measure for the estimated depth. As shown in Fig. 4, $\Lambda$ is typically lower on object boundaries—indicating higher uncertainty—and higher near sparse point measurements, where confidence is greater.

To efficiently compute the depth distribution, GBPN employs a Serial & Parallel propagation scheme. The ablation study (see Section 4.2 and Table 2) validates the efficacy of this design. For instance,

introducing serial propagation (comparing $V_2$ and $V_3$) resulted in a substantial performance gain, reducing RMSE from $340.84mm$ to $108.29mm$. Furthermore, augmenting the model with parallel propagation over non-local edges (comparing $V_6$ and $V_7$) yielded a further improvement, lowering RMSE from $103.92mm$ to $101.42mm$. In addition to these quantitative results, we provide the qualitative analysis through visualizations to illustrate the impact of the propagation scheme.

To validate the information propagation capability of our serial propagation scheme, we modified GBPN by removing dynamic parameters and edges, resulting in a model that uses only four-directional (left-to-right, top-to-bottom, right-to-left, bottom-to-top) serial propagation. When trained on the NYUv2 dataset with 500 sparse points and tested on an extreme case with only a single depth measurement, the model successfully propagates information across the entire image within a single iteration. The intermediate results, illustrated in Fig. 5, demonstrate rapid convergence to a meaningful depth map, confirming the scheme's effectiveness.

Unlike traditional MRFs with static, predefined graph edges (e.g., an 8-connected grid), our framework enhances the graph structure by inferring dynamic, non-local edges for each pixel, as illustrated in Fig. 6. These edges connect contextually relevant pixels beyond immediate spatial neighborhoods, enabling the MRF to capture long-range dependencies and model complex scene structures more effectively. GBPN uses a parallel propagation scheme to pass messages simultaneously along these non-local connections, thereby conditioning each variable on both its local neighbors and its learned, non-local context.

It is worth noting that while the theoretical computational complexity of parallel propagation is similar to that of serial propagation (and significantly less than the cost of generating the MRF in GMCN, as detailed in Section A.7), its practical implementation is far more efficient. The parallel scheme readily leverages modern hardware parallelism, resulting in faster execution.

### A.4 MORE COMPARISON WITH STATE-OF-THE-ART (SOTA) METHODS

The comprehensive comparison of various depth completion methods on the KITTI and NYUv2 datasets is presented in Table 5. The results reveal that the proposed GBPN method achieves competitive performance across all metrics. On KITTI, it achieves the best iRMSE and second-best RMSE, indicating strong depth accuracy and consistency. On NYUv2, it matches the lowest RMSE and achieves the highest $\delta_{1.02}$ and $\delta_{1.05}$, reflecting excellent depth accuracy.

Visual comparisons with other SOTA open-source methods on the validation set of KITTI DC and the test set of NYUv2 are shown in Fig. 7 and Fig. 8. The official implementations and the best-performing models released by the authors are used to ensure fair comparisons, with the same sparse depth maps across all methods. Our results, presented in the last row, demonstrate sharper object boundaries and more detailed structures, while other methods tend to underperform in these challenging areas, leading to less accurate depth estimates.

### A.5 MORE COMPARISON ON SPARSITY ROBUSTNESS

### A.5.1 SIMULATION WITH REDUCED LIDAR SCAN LINES

To further evaluate our model's performance on sparser inputs, we tested all compared methods using sub-sampled LiDAR data corresponding to 8, 16, 32, and 64 scan lines. A detailed comparison on the KITTI validation set is shown in Table 6. Across all levels of sparsity, our method consistently delivers the best performance for RMSE, REL, and MAE. Notably, under the most challenging 8-line setting, our model demonstrates a clear advantage over other methods. For example, compared to BP-Net, GBPN reduces the RMSE from 4541.9 mm to 2750.4 mm, yielding a substantial 1791.5 mm absolute improvement. This underscores the robustness and adaptability of our method, particularly in scenarios with extremely sparse depth input. Visual results in Fig. 9 further support these findings. Our model effectively preserves structural details even with limited LiDAR information, whereas competing methods tend to produce smoother or more ambiguous results with fewer LiDAR lines.

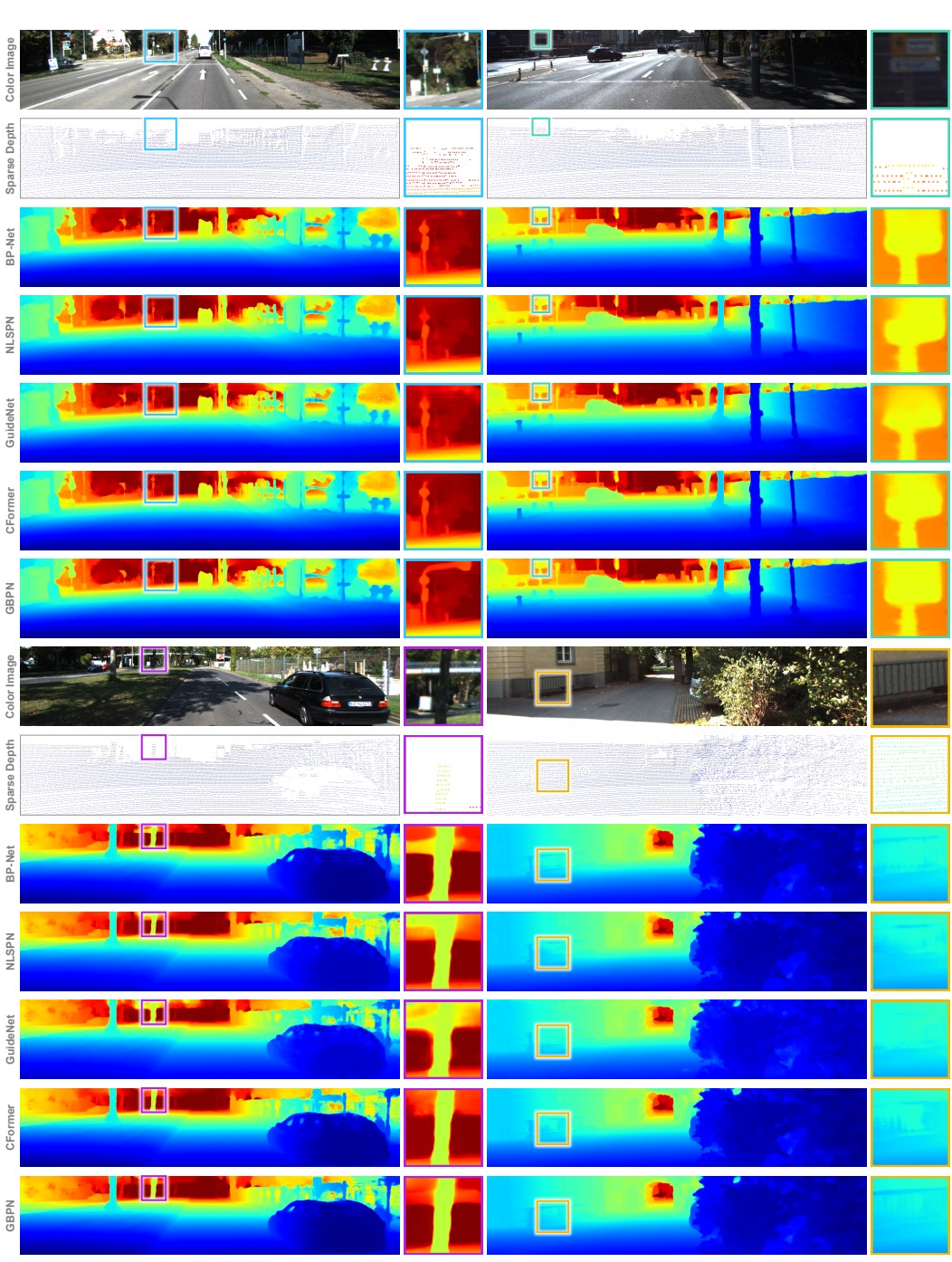

Figure 7: **Qualitative comparison on KITTI validation set.** Comparing with GuideNet (Tang et al., 2020), NLSPN (Park et al., 2020), CFormer (Zhang et al., 2023) and BP-Net (Tang et al., 2024). Our method is presented in the last row, with key regions highlighted by rectangles for easy comparison

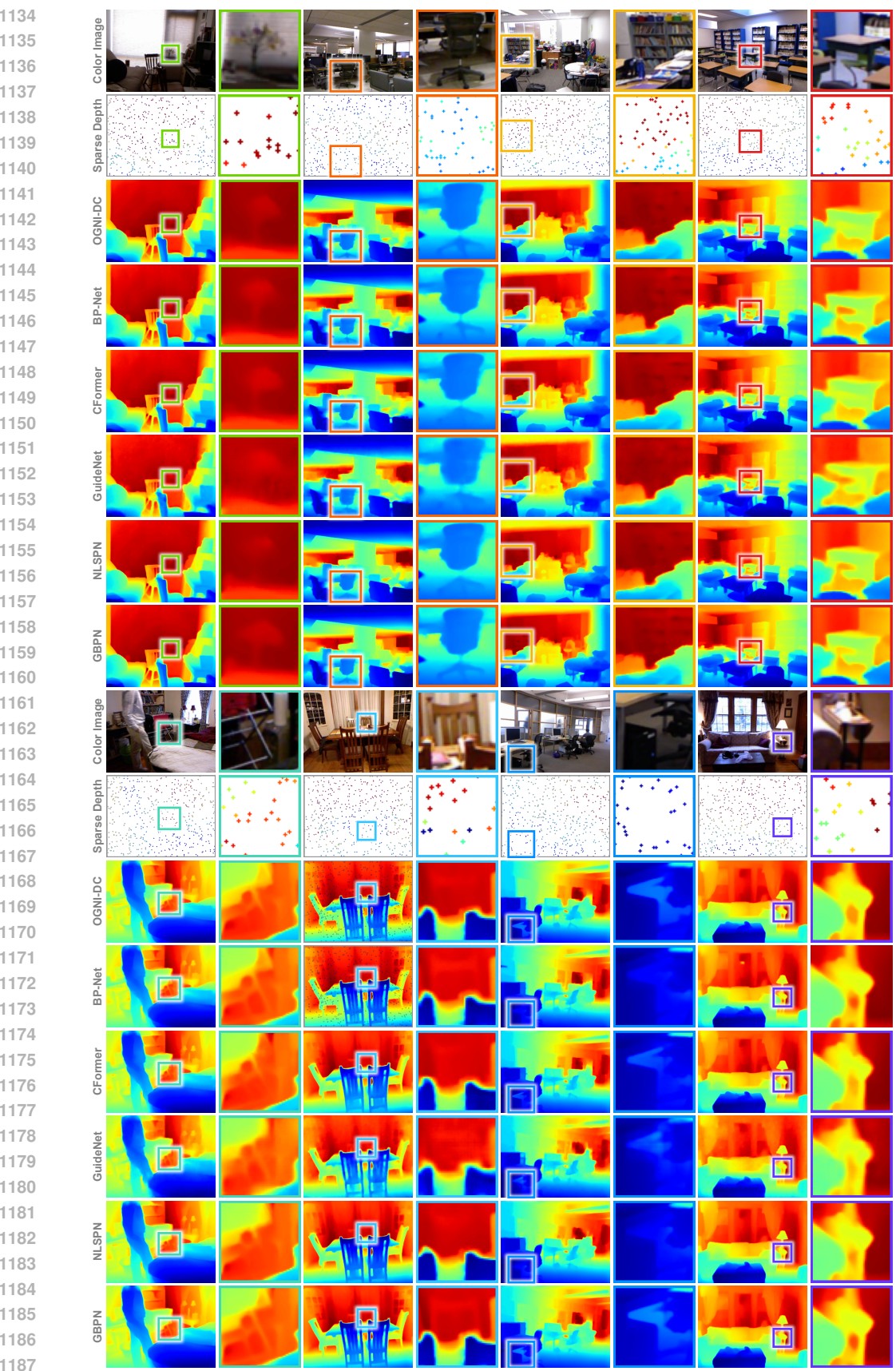

Figure 8: **Qualitative comparison on NYUv2 test set.** Our method is compared with BP-Net (Tang et al., 2024), NLSPN (Park et al., 2020), GuideNet (Tang et al., 2020), CFormer (Zhang et al., 2023) and OGNI-DC (Zuo & Deng, 2024). For clearer visualization, sparse depth points are enlarged. Our method is presented in the last row, with key regions highlighted by rectangles to facilitate comparison.

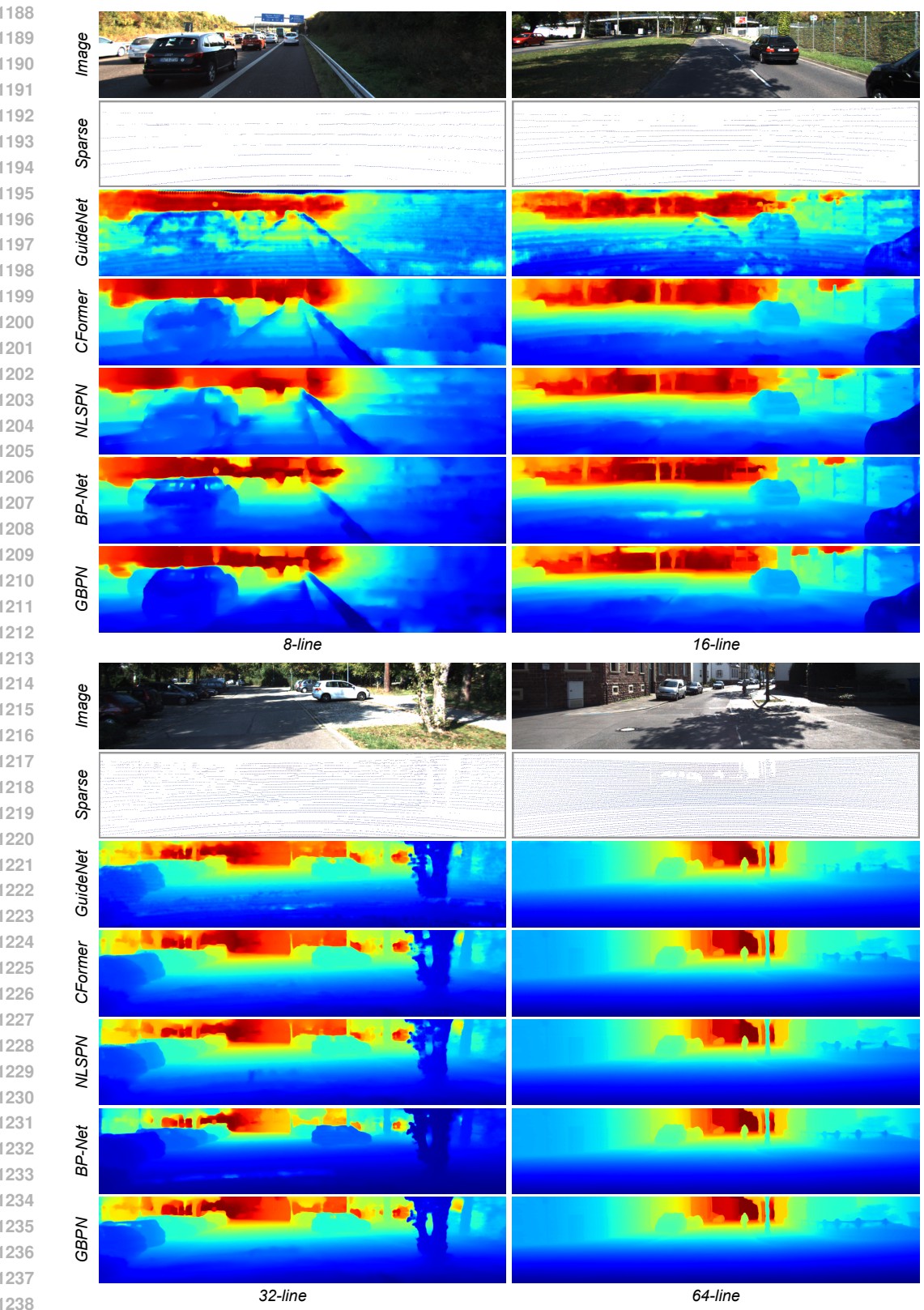

Figure 9: **Qualitative comparison on KITTI validation set with different LiDAR line.** Our method is compared with BP-Net (Tang et al., 2024), NLSPN (Park et al., 2020), GuideNet (Tang et al., 2020) and CFormer (Zhang et al., 2023).

Table 5: **Performance on KITTI and NYUv2 datasets.** For the KITTI dataset, results are evaluated by the KITTI testing server. For the NYUv2 dataset, authors report their results in their papers. The best result under each criterion is in **bold**. The second best is with underline.

| | KITTI | | | | NYUv2 | | | |
|---|---|---|---|---|---|---|---|---|
| | RMSE↓ (mm) | MAE↓ (mm) | iRMSE↓ (1/km) | iMAE↓ (1/km) | RMSE↓ (m) | REL↓ | $\delta_{1.02}$↑ (%) | $\delta_{1.05}$↑ (%) |
| S2D (Ma & Karaman, 2018) | 814.73 | 249.95 | 2.80 | 1.21 | 0.230 | 0.044 | – | – |
| DeepLiDAR (Qiu et al., 2019) | 758.38 | 226.50 | 2.56 | 1.15 | 0.115 | 0.022 | – | – |
| CSPN++ (Cheng et al., 2020) | 743.69 | 209.28 | 2.07 | 0.90 | 0.115 | – | – | – |
| GuideNet (Tang et al., 2020) | 736.24 | 218.83 | 2.25 | 0.99 | 0.101 | 0.015 | 82.0 | 93.9 |
| FCFR Hu et al. (2021) | 735.81 | 217.15 | 2.20 | 0.98 | 0.106 | 0.015 | – | – |
| NLSPN (Park et al., 2020) | 741.68 | 199.59 | 1.99 | 0.84 | 0.092 | 0.012 | 88.0 | 95.4 |
| ACMNet (Zhao et al., 2021a) | 744.91 | 206.09 | 2.08 | 0.90 | 0.105 | 0.015 | – | – |
| RigNet Yan et al. (2022) | 712.66 | 203.25 | 2.08 | 0.90 | 0.090 | 0.013 | – | – |
| DySPN (Lin et al., 2022) | 709.12 | 192.71 | 1.88 | 0.82 | 0.090 | 0.012 | – | – |
| BEV@DC (Zhou et al., 2023) | 697.44 | 189.44 | 1.83 | 0.82 | 0.089 | 0.012 | – | – |
| DGDF (Wang et al., 2023b) | 707.93 | 205.11 | 2.05 | 0.91 | 0.098 | 0.014 | – | – |
| CFormer (Zhang et al., 2023) | 708.87 | 203.45 | 2.01 | 0.88 | 0.090 | 0.012 | 87.5 | 95.3 |
| LRRU (Wang et al., 2023a) | 696.51 | 189.96 | 1.87 | **0.81** | 0.091 | 0.011 | – | – |
| TPVD (Yan et al., 2024) | 693.97 | 188.60 | 1.82 | **0.81** | 0.086 | **0.010** | – | – |
| ImprovingDC (Wang et al., 2024) | 686.46 | **187.95** | 1.83 | **0.81** | 0.091 | 0.011 | – | – |
| OGNI-DC (Zuo & Deng, 2024) | 708.38 | 193.20 | 1.86 | 0.83 | 0.087 | 0.011 | 88.3 | 95.6 |
| BP-Net (Tang et al., 2024) | 684.90 | 194.69 | 1.82 | 0.84 | 0.089 | 0.012 | 87.2 | 95.3 |
| DMD3C (Liang et al., 2025) | **678.12** | 194.46 | 1.82 | 0.85 | **0.085** | 0.011 | – | – |
| GBPN | 682.20 | 192.14 | **1.78** | 0.82 | **0.085** | 0.011 | **89.1** | **95.9** |

Table 6: Performance on KITTI's validation set with different simulated lines. All compared methods are tested using sub-sampled LiDAR data with 8, 16, 32, and 64 scan lines.

| LiDAR Scans | 8-Line | | | 16-Line | | | 32-Line | | | 64-Line | | |
|---|---|---|---|---|---|---|---|---|---|---|---|---|
| Methods | RMSE | REL | MAE | RMSE | REL | MAE | RMSE | REL | MAE | RMSE | REL | MAE |
| CFormer (Zhang et al., 2023) | 3660.5 | 106.4 | 1720.3 | 2196.2 | 46.4 | 822.1 | 1242.2 | 21.9 | 380.9 | 745.6 | 10.7 | 197.3 |
| NLSPN (Park et al., 2020) | 3244.2 | 92.3 | 1512.2 | 1949.8 | 38.4 | 676.2 | 1195.2 | 20.5 | 353.5 | 771.8 | 10.5 | 197.3 |
| BP-Net (Tang et al., 2024) | 4541.9 | 143.8 | 1822.8 | 2362.7 | 52.0 | 1822.8 | 1425.2 | 29.6 | 399.6 | 715.2 | 10.4 | 194.0 |
| GuideNet (Tang et al., 2020) | 5805.8 | 216.2 | 3109.2 | 3291.6 | 93.5 | 1455.8 | 1357.5 | 28.5 | 452.7 | 770.8 | 12.3 | 222.2 |
| GBPN | **2750.4** | **79.5** | **1233.6** | **1744.1** | **31.6** | **560.6** | **1073.1** | **17.6** | **311.4** | **712.4** | **10.3** | **191.8** |

### A.5.2 Simulation with Various Depth Density

The metrics on the NYUv2 validation set under various levels of input depth sparsity are presented in Table 7, ranging from 20 to 20,000 sparse points. For a thorough evaluation, given a sparsity level, each test image is sampled 100 times with different random seeds to generate the input sparse depth map. The performance of each method is averaged on these 100 randomly sampled inputs to reduce the potential bias due to random sampling, especially for high sparsity.

Table 7: RMSE (m) on NYUv2 with depth input from various sparsity. The best metric under each sparsity level is in **bold**.

| Points | 20 | 50 | 100 | 200 | 500 | 1000 | 2000 | 5000 | 10000 | 20000 |
|---|---|---|---|---|---|---|---|---|---|---|
| CFormer (Zhang et al., 2023) | 0.932 | 0.709 | 0.434 | 0.142 | 0.091 | 0.071 | 0.056 | 0.045 | 0.066 | 0.204 |
| NLSPN (Park et al., 2020) | 0.691 | 0.429 | 0.248 | 0.136 | 0.092 | 0.072 | 0.056 | 0.042 | 0.035 | 0.035 |
| OGNI-DC (Zuo & Deng, 2024) | 1.523 | 1.149 | 0.696 | 0.172 | 0.088 | 0.069 | 0.054 | 0.040 | 0.032 | 0.030 |
| BP-Net (Tang et al., 2024) | 0.749 | 0.547 | 0.302 | 0.131 | 0.090 | 0.070 | 0.054 | **0.039** | 0.031 | **0.023** |
| GuideNet (Tang et al., 2020) | 0.908 | 0.603 | 0.478 | 0.187 | 0.101 | 0.081 | 0.070 | 0.087 | 0.195 | — |
| GBPN-1 | **0.647** | **0.331** | **0.179** | 0.135 | 0.101 | 0.081 | 0.065 | 0.048 | 0.037 | 0.027 |
| GBPN-2 | 0.649 | 0.364 | 0.198 | **0.120** | **0.085** | **0.067** | **0.053** | **0.039** | 0.032 | 0.026 |

Our method is listed in the last two rows, GBPN (with variants GBPN-1 and GBPN-2), consistently achieves the highly competitive performance across all sparsity levels. Under extremely sparse input, 20 and 50 points, GBPN-1 achieves the lowest RMSE, significantly outperforming other methods. As the number of input points increases, GBPN-2 starts to excel, obtaining the best performance at intermediate sparsity levels more than 100 points. Notably, both GBPN variants remain highly effective even at dense input levels (5000 and 20,000 points). In contrast, GuideNet (Tang et al., 2020) and Cformer (Zhang et al., 2023) has a worse performance when the input depth is significantly denser (beyond approximately 5000 points) than the training sparsity (500 points). Similar phenomenon has also been observed by (Zuo & Deng, 2024), and we attribute this to the lack of robustness to changes in input sparsity. Both GuideNet (Tang et al., 2020) and CFormer (Zhang et al., 2023) directly process the sparse depth map using convolutional layers, which are not optimal for handling sparse data. In addition, these methods were trained exclusively with 500 valid points. When presented with a significantly denser input at test time, the input distributions passed to the initial convolutional layers differ drastically from what the network was trained on. This domain shift may cause the network to produce worse results, regardless of whether the input density is increased or decreased. Our method alleviate this issue by treating sparse depth measurements as principled observation terms within a globally optimized MRF framework.

The qualitative comparison in Fig. 10 further confirms these findings. Despite varying sparsity levels, our method reconstructs sharper structures and more accurate depth boundaries compared to existing approaches. In particular, it maintains strong performance under sparse input conditions where other methods tend to blur edges or produce overly smooth estimations. These results demonstrate that GBPN is both robust and generalizable across a wide range of input sparsity, making it well-suited for real-world scenarios with depth measurements of various sparsity, *e.g.* SfM (Schonberger & Frahm, 2016).

## A.6 NOISE SENSITIVITY

In the real world, depth measurements may tend to be noisy due to sensor errors or environmental factors like fog. To evaluate the noise sensitivity of DC methods, we evaluate the compared methods with noisy depth input. Specifically, each sparse depth measurement is perturbed by a uniformly distributed relative error, $\theta$. For instance, $\theta = 1\%$ indicates an error uniformly sampled between $-1\%$ and $1\%$ of the true depth value. All models were trained on the clean NYUv2 dataset and evaluated on these corrupted inputs. As shown in Table 8, GBPN consistently achieves the best RMSE across noise levels ranging from $\theta = 1\%$ to $\theta = 5\%$, demonstrating its robustness against noise and its ability to maintain stable and accurate depth predictions even under corrupted measurements.

Table 8: RMSE (mm) Comparison across different levels of noise in sparse depth measurements.

| Method | $\theta = 1\%$ | $\theta = 2\%$ | $\theta = 3\%$ | $\theta = 4\%$ | $\theta = 5\%$ |
|---|---|---|---|---|---|
| OGNI (Zuo & Deng, 2024) | 89.82 | 93.31 | 98.77 | 107.11 | 116.23 |
| NLSPN (Park et al., 2020) | 93.48 | 96.86 | 101.93 | 110.38 | 119.72 |
| CFormer (Zhang et al., 2023) | 92.81 | 96.27 | 101.88 | 110.12 | 119.98 |
| BP-Net (Tang et al., 2024) | 90.63 | 93.54 | 98.43 | 106.12 | 113.85 |
| GuideNet (Tang et al., 2020) | 101.91 | 104.12 | 108.32 | 114.92 | 122.51 |
| GBPN | **87.17** | **90.41** | **95.60** | **104.01** | **113.10** |

## A.7 EFFICIENCY COMPARISON

We compare the runtime efficiency of DC methods on samples from NYU and KITTI dataset with input resolution of $256 \times 320$ and $352 \times 1216$, respectively. As shown in Table 9, GBPN achieves a competitive balance of runtime, memory usage, and parameter count compared with these SOTA methods. Here, GuideNet demonstrates the fastest inference speed, while CFormer and OGNI-DC are notably slower. Our GBPN exhibits a moderate inference time compared to these SOTA methods. Comparing with BP-Net(Tang et al., 2024), GBPN has a higher runtime but requires fewer parameters. The lower parameter count is because the Markov Random Field (MRF) is dynamically constructed by a relatively small network compared to other methods.

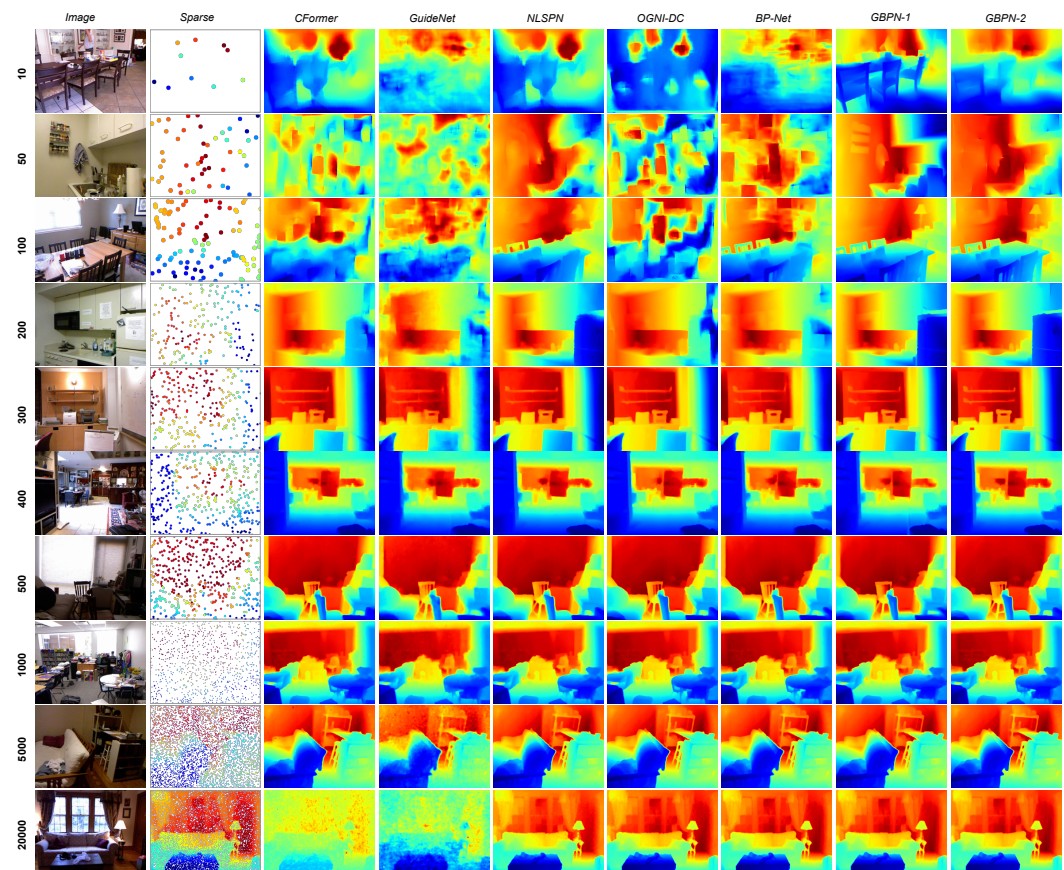

Figure 10: **Qualitative comparison on NYUv2 test set with different sparsity.** Our method is compared with BP-Net (Tang et al., 2024), NLSPN (Park et al., 2020), GuideNet (Tang et al., 2020), CFormer (Zhang et al., 2023) and OGNI-DC (Zuo & Deng, 2024). For each sparsity level, the first row is input image, the second row is sparse map.

Table 9: Comparison of inference time (ms), GPU memory (GB), and number of parameters (M).

| Method | 256 × 320 | | 352 × 1216 | | Params (M) |
| --- | --- | --- | --- | --- | --- |
| | Runtime (ms) | GPU Memory (GB) | Runtime (ms) | GPU Memory (GB) | |
| GuideNet (Tang et al., 2020) | 3.52 | 4.65 | 12.95 | 8.85 | 62.62 |
| NLSPN (Park et al., 2020) | 8.50 | 1.26 | 39.34 | 5.69 | 26.23 |
| BP-Net (Tang et al., 2024) | 19.87 | 5.31 | 77.64 | 9.07 | 89.87 |
| CFormer (Zhang et al., 2023) | 115.50 | 1.98 | 150.76 | 7.85 | 83.51 |
| OGNI-DC (Zuo & Deng, 2024) | 177.02 | 2.77 | 266.06 | 9.25 | 84.37 |
| GBPN | 44.57 | 4.47 | 137.49 | 8.70 | 39.03 |

Our GBPN consists of a Graphical Model Construction Network (GMCN) to dynamically construct a scene-specific Markov Random Field (MRF), and a Gaussian Belief Propagation (GBP) module for depth distribution inference. We analyzed the computation and runtime of these two components at an input resolution of $256 \times 320$. The GMCN is built using highly-optimized PyTorch layers, whereas our GBP module is a custom implementation. As shown in Table 10, the computation of the GBP module are roughly 200 times lower than those of the GMCN, yet it requires 3 times more computation time. This inefficiency, where less computation takes more time, is primarily due to the limitations of our custom GBP implementation. Although our current implementation is parallelized

on GPUs, it lacks the extensive optimization like the standard PyTorch layers and thus has significant potential for acceleration in future work.

Table 10: Comparison of computation and runtime between GMCN and GBP.

| Components | Computation (GFLOPs) | Runtime (ms) |
|---|---|---|
| GMCN | 62.58 | 11.03 |
| GBP | 0.26 | 33.54 |

Our GBPN also has the advantage of *flexible latency tuning*. As the target depth is optimized gradually in an iterative manner, we can trade off the model's latency and performance by simply adjusting the number of iterations. We list the trade-off in Table 11 by starting from our GBPN model with a total of 13 iterations and then gradually reducing the number of iterations. This property is highly beneficial for real applications' deployment, as GBPN can be easily tailored to specific latency requirements by simply adjusting the number of optimization iterations. In contrast, feed-forward networks like CFormer (Zhang et al., 2023) must complete their full inference pass to produce the result, making them difficult to adapt to varying latency budgets.

Table 11: Trade-off between latency and accuracy on the NYUv2 validation set under different number of iterations.

| Iteration | 5 | 6 | 7 | 8 | 9 | 10 | 11 | 12 | 13 |
|---|---|---|---|---|---|---|---|---|---|
| RMSE under 500 points (m) | 0.146 | 0.141 | 0.117 | 0.106 | 0.099 | 0.098 | 0.090 | 0.087 | 0.085 |
| RMSE under 400 points (m) | 0.157 | 0.151 | 0.126 | 0.114 | 0.107 | 0.104 | 0.096 | 0.093 | 0.091 |
| RMSE under 300 points (m) | 0.178 | 0.171 | 0.140 | 0.128 | 0.119 | 0.113 | 0.104 | 0.101 | 0.100 |
| RMSE under 200 points (m) | 0.241 | 0.228 | 0.184 | 0.173 | 0.154 | 0.137 | 0.124 | 0.121 | 0.119 |
| RMSE under 100 points (m) | 0.471 | 0.439 | 0.356 | 0.344 | 0.252 | 0.298 | 0.238 | 0.233 | 0.214 |
| Runtime (ms) | 22.17 | 24.98 | 27.97 | 31.35 | 33.41 | 36.55 | 39.27 | 42.00 | 44.57 |

## A.8 DECLARATION OF LLM USAGE

This is an original research paper. The core method development in this research does not involve LLMs as any important, original, or non-standard components. LLM is used only for editing and formatting purposes.

