# OpenReview forum: "Gaussian Belief Propagation Network for Depth Completion"
_ICLR.cc/2026/Conference — Submitted to ICLR 2026_

### Official Review · Reviewer_rfq1 · 2025-10-29

**Soundness:** 3
**Presentation:** 3
**Contribution:** 2
**Rating:** 2
**Confidence:** 5

**Summary:**

This paper proposes the Gaussian Belief Propagation Network (GBPN) for depth completion. GBPN leverages a learned Markov Random Field (MRF) structure, constructed dynamically from RGB and sparse depth inputs, and performs inference via Gaussian Belief Propagation (GBP). The paper introduces a hybrid message-passing scheme and evaluates the method on NYUv2 and KITTI benchmarks, reporting competitive results.

**Strengths:**

1. Hybrid Learning-Inference Integration: The paper attempts a meaningful integration of graphical model reasoning with deep learning. Learning the MRF structure dynamically from images represents a conceptual advance over fixed priors or fully feed-forward architectures.

2. Principled Treatment of Sparse Inputs: The method embeds sparse depth directly into the probabilistic inference process, rather than processing it as part of standard CNN input, offering a more theoretically grounded approach to the sparsity challenge.

**Weaknesses:**

1. Lack of Computational Analysis: The proposed approach introduces significant inference overhead due to iterative GBP and dynamic graph construction. However, the paper does not report any runtime statistics, GPU memory usage, or scalability discussion. Given the growing importance of efficiency in practical systems, this omission is concerning.

2. Limited Empirical Gain: While the method shows SOTA iRMSE on KITTI, it underperforms on other key metrics (RMSE, MAE), suggesting the gain may not be consistent. On NYUv2, although the reported RMSE is strong, the deltas are small and the competitive landscape is already saturated.

3. Weak Justification of Components: Ablation studies show marginal gains (~3mm RMSE difference on NYUv2), raising questions about the necessity of the complex model components, including non-local edge prediction, dynamic updates, and dual-pass U-Nets.

4. Insufficient Analysis on Iterative Behavior: The authors claim that more iterations improve performance (line 290), but Table 2 only reports 3 and 5 iterations. It remains unclear whether the performance plateaus or continues to improve, and at what computational cost.

5. Unclear Sparsity Robustness Comparison: In Figure 2, the RMSE of some methods (e.g., GuideNet, CFormer) increases as input becomes denser, which is counter-intuitive and not explained. Additionally, curves for many methods converge from 500 points onward, making relative robustness claims less persuasive.

6. Presentation and Layout: Several pages are cluttered with tightly packed text and figures (notably pages 6 and 9), negatively impacting readability.

**Questions:**

A key concern is that the ablation results are unconvincing, as the reported gain is only about 3 mm. For the same model, it is quite common that retraining multiple times yields variations of around 5 mm, which means such a small improvement is either unsuitable for ablation analysis or insufficient to demonstrate the effectiveness of individual modules. It is also puzzling that the model’s accuracy degrades when the density of depth points increases, yet the explanations provided fail to address this anomaly satisfactorily.

Even after this issue was highlighted by the reviewers in **NeurIPS comments** before, the authors showed no intention of taking concrete steps to rigorously validate the effectiveness of their method or to improve its interpretability. The current revision remains insufficient for publication, and the reviewer maintains a negative overall assessment of the paper.

---

> ### Author Response · Authors · 2025-11-21
> **Response to Reviewer rfq1 [Part 1/3]**
>
> Thank you for your thorough review, We appreciate the opportunity to clarify these points. Below are our point-by-point responses.
>
> **Q1 \& W3**: The key concern is that ablation study is unconvincing: the improvement is only ~3 mm. For the same model, it is quite common that retraining multiple times yields variations of around 5 mm.
>
> >**A**: Thank you for this important question. We agree that a clear and convincing ablation study is crucial. In our revised version, we have completely restructured the ablation study to rigorously demonstrate the contribution of each component.
>
> >**Addressing the Variation Concern**: We respectfully provide some context regarding performance variation. On the NYUv2 benchmark, the community has achieved a performance gain of ~5 mm RMSE (from 90mm to 85mm) over several years, as documented in numerous papers (see our Table 1 & 5). A variation of 5mm from retraining would indeed invalidate these reported improvements. To directly address your concern, we performed three additional training runs of our GBPN-1 with different random seeds. The results were highly stable: 100.65 mm, 100.69 mm, and 100.74 mm RMSE (a variation of <0.1 mm). We attribute this stability to our use of the EMA training strategy (Line 324). Therefore, the performance gain of our core components is not attributable to training variance. **To avoid the potential confusion, we restructured the ablation study**.
>
> >**For revision**: We now present the ablation studies by introducing components incrementally, starting from a simple baseline ($V_1$) and progressively adding modules up to the final model ($V_9$). This approach ensures the configuration of each variant is explicit and the individual effectiveness of every component is clearly verifiable.
>
> >Specifically, we provide detailed configurations and analysis results starting from Line 411. We have also restructured Table 2 into four separate parts to facilitate a direct comparison of each core component's impact. For a comprehensive evaluation, we list both RMSE and the accuracy metric $\delta_{1.025}$. As shown in the updated results, moving from the baseline ($V_1$) to the final model ($V_9$), the RMSE drops from $342.70 mm$ to $100.69 mm$, and $\delta_{1.025}$ improves from $21.58\\%$ to $85.27\\%$. The large performance improvement demonstrates the strength and priority of our GBPN architecture and the effective contribution of each core component.

---

> ### Author Response · Authors · 2025-11-21
> **Response to Reviewer rfq1 [Part 2/3]**
>
> **Q2 \& W5**: It is also puzzling that the model’s accuracy degrades when the density of depth points increases, yet the explanations provided fail to address this anomaly satisfactorily.
>
>
> >**A**: Thank you for this important question. We have revised the manuscript to provide a more detailed explanation of this phenomenon to avoid confusing readers.
>
> >**Clarification**: We respectfully clarify that this phenomenon is not a weakness of our method, but rather reflects limitations in comparison methods like GuideNet and CFormer, where performance degrades at significantly higher input densities. We provide the support for **correctness** of this phenomenon and the corresponding **explanation** in the following.
>
>
> > **For correctness**: We'd like to kindly note this is indeed a known issue, and similar behavior has been reported in prior works (e.g., OGNI-DC (results listed in Table 3 and visualized in Fig.c)). Our observation is consistent with theirs. Moreover, as the code and model of these methods are released, it can be easily verified by simply a line command. For example, download the CFormer's code and model, and directly run 'python main.py --dir_data PATH_TO_NYUv2 --data_name NYU --gpus 0 --max_depth 10.0 --num_sample 500 --test_only'. It will shown with 500 points, the RMSE is 0.09013, and run 'python main.py --dir_data PATH_TO_NYUv2 --data_name NYU --gpus 0 --max_depth 10.0 --num_sample 20000 --test_only' with 20000 points, the RMSE is 0.20321. This is consistent with the Figure 3 and Table 7 reported in our paper. Therefore, the result we reported is correct.
>
> > **For explanation**: We believe this phenomenon reveals a lack of robustness to variations in input sparsity in some architectures. We posit this is a direct consequence of a train-test distribution shift. These methods typically adopt convolutional layers to directly process sparse depth image. As analyzed in BPNet, convolutional layer is not an optimal operator for sparse data. In addition, these methods were trained exclusively with 500 valid points. When presented with a significantly denser input at test time, the input distributions passed to the initial convolutional layers (e.g., in GuideNet) differ drastically from what the network was trained on. This domain shift may cause the network to produce worse results, regardless of whether the input density is increased or decreased.
>
> > **For revision**:  we have modify the section 4.3 in line 487 to make the analysis more clear. Also, we give the more detailed explanation in the section A.5.2 line 1301 to avoid the potential confusion.

---

> ### Author Response · Authors · 2025-11-21
> **Response to Reviewer rfq1 [Part 3/3]**
>
> **W1** : Lack of Computational Analysis
>
> >**A**: Thanks for your question, The computation analysis is in section A.7. We compare and analyze the efficiency with other methods and the computation and runtime of our GBPN. Also, we have refer this in the Experiment section line 371, "The appendix contains further analysis on the experimental setup (Section A.2), noise sensitivity (Section A.6), and runtime efficiency (Section A.7)."
>
>
> **W2** : Limited Empirical Gain
>
>
> > **A**: Thanks for your question. We'd like to respectfully note GBPN not only has better accuracy, but also better robustness and genelization capability. For real application, the robustness and genelization capability would be as same if not more important as accuracy. We have made this more clear to readers in the revised version.
>
> > **For performance**: GBPN delivers consistently strong results across both the KITTI and NYUv2 benchmarks. It outperform the strong competitor BP-Net among all the evaluation metrics (RMSE, MAE, REL, $\delta_{1.02}$, etc.) on these two leading benchmarks.  In particular, GBPN achieves a large improvement on NYU dataset with the best RMSE and $\delta_{1.02}$. As listed in Table 1, compared to BP-Net, the $\delta_{1.02}$ of GBPN improves from $87.2\\%$ to $89.1\\%$ (i.e. $1.9\\%$ absolute improvement). Tables 6 and 7 show that GBPN consistently outperforms all competing methods, achieving the lowest errors under each LiDAR line or point density setting.  This advantage becomes even more pronounced in extremely sparse regimes (e.g., 8-line KITTI and 20–100 points on NYUv2). As listed in Table 6, with depth measured by 8-line LiDAR on KITTI benchmark, compared to BP-Net, the RMSE of GBPN improves from $4541.9 mm$ to $2750.4 mm$ (i.e. $1791.5 mm$ absolute improvement).   As listed in Table 7, with 20 points depth measurement on NYUV2 benchmark, compared to BP-Net, the RMSE of GBPN improves from $0.749 m$ to $0.649 m$ (i.e. $0.1 m$ absolute improvements). Furthermore, with zero-shot cross dataset evaluation on VOID benchmark,  GBPN achieves the best RMSE and MAE across all sparsity levels , demonstrating consistently stronger generalization capability across sparsity, pattern, and scenes. As listed in Table 3, with 1500 sparse depth from visual odometry system on VOID benchmark, compared to BP-Net, the RMSE of GBPN improves from $0.74 m$ to $0.68 m$ (i.e. $0.06 m$ absolute improvement). In summary, considering the accuracy, robustness and genelization capability, the performance improvement of GBPN is not marginal compared to any competitor.
>
> > **For revision**: We have revised the Abstract section (line 026), Introduction section (line 071) to highlight the substantial performance improvement of GBPN. we change the section about generalizable capability from appendix to the main paper(section 4.4) to make the main paper self-contain. We add more analysis on the computation and latency in section A.7 (line 1390) and Table 11. Promising to optimize the implementation in Conclusion section (line 538).
>
>
> **W4**: Insufficient Analysis on Iterative Behavior
>
> >**A**: Thanks for your question. We experimented with the number of iterations, starting from 1 and increasing it incrementally. We observed that performance improves as the number of iterations increases. While, beyond 5 iterations, the performance gain becomes negligible.
>
> > **For revision**: We analyzed the impact of iterative in line 454.
>
>
>
> **W6**: Presentation and Layout
> >**A**: Thanks for your question. As suggested, we have revised the presentation and layout to improve the readability.
>
>
>
>
> Best Regards,
>
> Authors of submission 18865

---

> > ### Comment · Reviewer_rfq1 · 2025-11-27
> >
> > Thanks for the feedback. Please revise the paper in accordance with the reviewers’ comments rather than disregarding those valuable suggestions. I am inclined to maintain my initial rating.

---

> > > ### Author Response · Authors · 2025-11-27
> > > **Response to Reviewer rfq1**
> > >
> > > Thank you for your feedback. We have carefully considered all your comments and have revised our manuscript accordingly. In our point-by-point response, we addressed all the concerns you raised, including the six weaknesses and two questions. We would be very grateful if you could specify which suggestions you feel were disregarded, as we are committed to incorporating all valuable feedback to improve our work. Thank you for your time and consideration.
> > >
> > > Best Regards,
> > >
> > > Authors of submission 18865

---

> > > > ### Comment · Reviewer_rfq1 · 2025-11-27
> > > >
> > > > Thank you for your efforts.
> > > >
> > > > In fact, when I first read your ICLR submission (resubmitted from NeurIPS), I was quite shocked, as most of the comments raised during the NeurIPS review were not reflected in the revised manuscript. Although you have now responded to my points one by one and made changes accordingly, this is not sufficient to turn an initially negative score into a strongly positive one. Nevertheless, your effort is evident, and therefore I have decided to raise the score from 2 to 4.
> > > >
> > > > Additionally, I am not sure whether your co-authors or advisors are aware of the NeurIPS review comments, and more importantly, whether they realize that your resubmitted version has not incorporated any of those changes. I believe that your senior advisors would, to some extent, have encouraged you to revise the resubmission if they are aware of the previous comments.
> > > >
> > > > I truly hope future versions can make further progress. Best wishes.

---

> > > > > ### Author Response · Authors · 2025-11-28
> > > > > **Response to Reviewer rfq1**
> > > > >
> > > > > Thank you for recognizing our efforts and for your valuable comments. We sincerely apologize for any impression that we did not adequately address the valuable feedback from the NeurIPS review. We would like to respectfully clarify that we did incorporate revisions for many of the concerns in our initial ICLR submission. Due to strict page limitations, most of these revisions—along with detailed discussions—are included in the appendix (e.g., Computational Analysis, Analysis of Iterative Behavior, and Analysis of Robustness Comparison). We fully understand that reviewers are not obliged to read the appendix, and we take responsibility for not making these integrations more explicit. In our revised version, we have taken significant steps to rectify this. We expanded our submission from 9 pages to 10 pages to include more analysis in the main paper, added explicit cross-references from the main text to the appendix, and conducted extensive new experiments (running several days) to provide a comprehensive and clear ablation analysis, which was your primary concern. Consequently, with the valuable suggestions from the rebuttal and discussions with all reviewers, our submission has been substantially enhanced. We hope this revised version convincingly addresses your concerns and demonstrates the progress we have made. We would be delighted to respond to any further concerns you may have, to enhance our work sufficiently to warrant a higher score.
> > > > >
> > > > > Best Regards,
> > > > >
> > > > > Authors of submission 18865

---

### Official Review · Reviewer_NRg4 · 2025-10-31

**Soundness:** 3
**Presentation:** 3
**Contribution:** 2
**Rating:** 6
**Confidence:** 3

**Summary:**

This paper addressed the depth completion task by developing the Gaussian Belief Propagation Network (GBPN). The GBPN consists of  a Graphical Model Construction Network (GMPN) for constructing a
scene-specific MRF over dense depth variables and the Gaussian
Belief Propagation strategy that infers the dense depth on the learned MRF. The GMPN models the potentials of the MRF and its structure by predicting
non-local edges to capture the  complex, long-range spatial dependencies
guided by image content. The GBP strategy uses serial & parallel message passing scheme to enhance information flow.

Experiments on KITTI and NYUv2 show that the proposed method achieves SOTA performance. The authors also conduct comprehensive ablations to validate the effectiveness of the proposed modules and the robustness.

**Strengths:**

The proposed method achieves SOTA performance on public benchmarks, KITTI and NYUv2

The authors validate the effectiveness of the proposed modules and the robustness over sparsity.

The authors also provide detailed information regarding the method, like model structure, proof, parameters, etc.

The idea of using use Gaussian Belief Propagation for inference is interesting, with strong motivation from previous methods.

**Weaknesses:**

- The strategy, MRF for depth estimation, has been explored before [1][2]. The authors should provide some discussions.

- What are the advantages of the MRF for depth completion (GMCN & GBP) in comparison with previous propagation-based methods?

- The authors claim that " allowing the model to adaptively capture complex, long-range spatial dependencies
guided by image content". Maybe it would be better if some cases are provided.

- As shown in Tab. 9 of the Supplementary, the proposed method has higher running time than BP-Net, while there performance is very close. Therefore, what advantages does the method have in comparison with BP-Net (apart from fewer parameters)

- Discussion about the Serial & Parallel Propagation Scheme should be provided, like the efficacy. How the strategy improves the performance/computational cost?

- In which scenarios, the proposed method performs better? and what issues it can solve? Please give more examples and analysis. Since the performance is close to the latest methods, the authors should give more evidences for the effectiveness of the proposed method.


[1] Chen et al., Fast MRF Optimization with Application to Depth Reconstruction.

[2] Liu et al., Deep Convolutional Neural Fields for Depth Estimation from a Single Image

**Questions:**

See the weakness. Please give feedbacks for each point.

---

> ### Author Response · Authors · 2025-11-21
> **Response to Reviewer NRg4 [Part 1/3]**
>
> Thank you for recognizing our work and for your valuable feedback. We have carefully considered your comments and address your concerns below.
>
> **W1**: The strategy, MRF for depth estimation, has been explored before [1][2]. The authors should provide some discussions.
>
> >**A**: Thank you for pointing this out. We agree that [1] and [2] are pioneering works utilizing MRF/CRF models for depth estimation. Our method shares the high-level motivation of leveraging probabilistic graphical models, but differs fundamentally in both problem setting and methodology.
>
> > 1. Problem Setting: The prior works [1, 2] address monocular depth estimation from a single image. In contrast, our work tackles the depth completion problem, which takes an RGB image coupled with sparse depth measurements as input. A central challenge we address is the sparse data processing and the effective information propagation. This is a specific core issue in depth completion that is not present in the monocular setting of [1, 2].
>
> > 2. Methodology: The methods in [1, 2] are designed to solve a pre-defined MRF/CRF with hand-crafted or fixed parameters and structure. A key innovation of our GBPN is that both the parameters and the graph structure of the MRF are dynamically inferred by the network, conditioned on the input. This data-driven approach yields a much more powerful and adaptive graphical model for capturing complex scene structures.
>
> > **For Revision**: As suggested, we have added discussions of these two works in the related work section. We discuss work [1] in line 92 and work [2] in line 113.
>
> **W2**: What are the advantages of the MRF for depth completion (GMCN & GBP) in comparison with previous propagation-based methods?
>
>
> >**A**: We thank the reviewer for raising this important point regarding the conceptual distinctions between our GBPN and other propagation-based methods. We appreciate the opportunity to provide a detailed explanation. We will first elaborate on the key differences and then provide a summary.
>
> >**Explaining in detail**: Firstly, existing propagation-based methods still face challenges in processing sparse data. Originally introduced by CSPN, propagation-based methods for depth completion, including their variants like CSPN++, DyPN, and NLSPN, primarily function as post-processing refinement modules. Consequently, they require an initial depth estimate from a deep network, which itself faces the challenge in processing sparse data. BP-Net attempts to alleviate this by propagating depth in a pre-processing stage. However, its use of a small MLP for propagation yields insufficiently accurate results, necessitating additional modules like Multi-modal Fusion Network and CSPN-based post-processing for further refinement. In contrast, our GBPN introduces a unified framework that synergistically integrates deep learning with probabilistic graphical models. Sparse depth measurements are naturally incorporated as principled data terms within a globally consistent Markov Random Field (MRF), inherently addressing input sparsity and irregularity without requiring auxiliary modules.
>
>
> > Secondly, current propagation-based methods suffer from a limited information propagation range. Whether using local convolutions (as in CSPN variants) or nearest-neighbor MLPs (as in BP-Net), the propagation process is confined to local regions. Sparse depth information cannot reliably propagate across the entire image. For instance, in BP-Net, a pixel can only receive information from its four nearest depth measurements. In contrast, our GBPN guarantees that the depth at any pixel can influence, and be influenced by, information from the entire image space in just one iteration. The relationship between GBPN and other propagation-based methods is analogous to that of Transformers and CNNs: just as Transformers overcome the limited receptive field of CNNs in feature extraction, GBPN overcomes the limited propagation range in depth completion.
>
>
> >Thirdly, while typical propagation-based methods yield only a dense depth map (a point estimate in statistics), our GBPN estimates a full depth distribution, providing both a depth value and a confidence measure for each pixel. As introduced in line 058, the confidence map is highly valuable for downstream, risk-aware applications such as robotic planning.
>
>
> >**In summary**: Other propagation-based methods are limited by local propagation ranges and typically operate in a single stage (either pre- or post-processing), requiring combination with other modules to complete the task. In contrast, our GBPN is a unified framework for depth distribution estimation. It achieves global information propagation, allowing depth from any pixel to inform the entire image, without relying on other components.
>
> >**For Revision**:  we have revised the introduction section(line 054) to highlight the advantages. And emphasis the issue we solved compared to previous methods in related work section (line 124).

---

> ### Author Response · Authors · 2025-11-21
> **Response to Reviewer NRg4 [Part 2/3]**
>
> **W3**: The authors claim that "allowing the model to adaptively capture complex, long-range spatial dependencies guided by image content". Maybe it would be better if some cases are provided.
>
> >**A**: Thanks for your suggestion. As suggested, we had some cases to make it clear for our claim that "allowing the model to adaptively capture complex, long-range spatial dependencies guided by image content". We visualize the "adaptively capture complex, long-range spatial dependencies" in Figure 6 and analyzed in line 1032.
>
> > **For revision**: Besides the above cases, we also add legend of non-local edges in Figure 1 and detailed explanation of the non-local propagation in line 273.
>
> **W4**: The proposed method has higher running time than BP-Net, what advantages does the method have in comparison?
>
>
> > **A**: Thanks for your question. The advantages of GBPN over BP-Net are twofold: conceptual (detailed in **W2**) and empirical. We wish to highlight that GBPN improves not only in accuracy but also significantly in robustness and generalization capability, which are as critical as accuracy, if not more, for real-world applications. Furthermore, the observed inference latency is not primarily due to increased computational demands, but rather stems from our current naive implementation. We acknowledge this issue and plan to address it through optimization implementation in the future work.
>
> > **For performance**: GBPN delivers consistently strong results across both the KITTI and NYUv2 benchmarks. It outperforms the strong competitor BP-Net on all evaluation metrics (RMSE, MAE, REL, $\delta_{1.02}$, etc.). Notably, GBPN achieves substantial improvement on the NYUv2 dataset, obtaining the best RMSE and $\delta_{1.02}$. As shown in Table 1, compared to BP-Net, the $\delta_{1.02}$ of GBPN improves from $87.2\\%$ to $89.1\\%$ (a $1.9\\%$ absolute gain). Tables 6 and 7 further demonstrate that GBPN consistently surpasses all competing methods, achieving the lowest errors under each LiDAR line or point density setting. This advantage is particularly pronounced in extremely sparse regimes (e.g., 8-line KITTI and 20–100 points on NYUv2). For instance, in Table 6, with 8-line LiDAR on KITTI, GBPN improves the RMSE from $4541.9 mm$ to $2750.4 mm$ (an absolute improvement of $1791.5 mm$). Similarly, in Table 7, with 20 points on NYUv2, GBPN improves the RMSE from $0.749 m$ to $0.649 m$ (a $0.1 m$ absolute gain). Moreover, in a zero-shot cross-dataset evaluation on the VOID benchmark, GBPN achieves the best RMSE and MAE across all sparsity levels, demonstrating superior generalization capability across varying sparsity level, sparse patterns, and scenes. As listed in Table 3, with 1500 sparse points on VOID, GBPN improves the RMSE from $0.74 m$ to $0.68 m$ (a $0.06 m$ absolute gain). In summary, considering the comprehensive gains in accuracy, robustness, and generalization, the performance improvement of GBPN is substantial and not marginal compared to any competitor.
>
> > **For latency**: As shown in Table 9, our method exhibits moderate latency among current depth completion methods. For instance, CFormer and OGNI both require more inference time than GBPN. We have also analyzed the runtime of GBPN in Section A.7. Our approach adopts a small, lightweight network as its backbone. However, as indicated in the same section, the optimization stage, while computationally less intensive than the deep network, occupies most of the runtime. This efficiency bottleneck is primarily due to our custom, naive implementation of Gaussian Belief Propagation (GBP). Compared to highly-optimized official PyTorch layers, our GBP implementation has significant room for improvement, and we are committed to optimizing it in future work.
>
>
> > **For Revision**: We have revised the Abstract section (line 026), Introduction section (line 071) to highlight the substantial performance improvement of GBPN. We change the section about generalizable capability from appendix to the main paper(section 4.4) to make the main paper self-contain. We add more analysis on the computation and latency in section A.7 (line 1390) and Table 11. We also promise to optimize the implementation in Conclusion section (line 538).

---

> ### Author Response · Authors · 2025-11-21
> **Response to Reviewer NRg4 [Part 3/3]**
>
> **W5**: Discussion about the Serial & Parallel Propagation Scheme should be provided, like the efficacy. How the strategy improves the performance/computational cost?
>
> >**A**:  Thank you for the question. As suggested, we have revised the section A.3  to discuss the efficacy and computation cost of Serial & Parallel Propagation Scheme with visualization. Also, add more about analysis on computation cost in the section A.7.
>
>
>
> **W6**: In which scenarios, the proposed method performs better? and what issues it can solve?
>
> > **A**: Thanks for your question. GBPN is particularly effective in scenarios where the testing conditions differ significantly from the training data—a common challenge in practice due to variations in sparsity level, sparsity pattern (depth captured from different methods like LiDAR or SFM), and scenes. In these scenarios, the priority of GBPN becomes more evident. Unlike other propagation-based approaches that struggle with sparse data processing and constrained local propagation ranges, GBPN naturally incorporates sparse measurements into a global MRF framework. This provides an inherent robustness to sparsity and irregularity, effectively solving these issues. The conceptual advantages has been detailed in our answer to **W2**. And the consequent performance improvement has been detailed in our answer to **W4**.
>
>
> > **For revision**: We have revised the manuscript to highlight the scenarios where GBPN excels, particularly under varying sparsity levels, sparsity patterns, and diverse scenes. Specifically, we modified the section 4.4 and appendix A.5, add analysis on GBPN’s improving robustness to sparsity, irregularity and generalization.
>
>
> Best Regards,
>
> Authors of submission 18865

---

> > ### Comment · Reviewer_NRg4 · 2025-11-27
> >
> > Thank the authors for their detailed responses. I am willing to keep my initial score.

---

> > > ### Author Response · Authors · 2025-11-28
> > > **Response to Reviewer NRg4**
> > >
> > > Thank you for recognizing our work and for your valuable feedback. We are grateful for your thorough evaluation of our paper and for your active engagement during the discussion period. We are appreciated our responses have addressed all of your concerns. With the valuable suggestions from the rebuttal and discussions with all reviewers, our submission has been substantially enhanced. We would be delighted to respond to any further concerns you may have, to enhance our work sufficiently to warrant a higher score.
> > >
> > > Best Regards,
> > >
> > > Authors of submission 18865

---

### Official Review · Reviewer_p3v7 · 2025-10-31

**Soundness:** 3
**Presentation:** 3
**Contribution:** 3
**Rating:** 6
**Confidence:** 3

**Summary:**

This paper addresses the depth completion task by introducing a Graphical Model Construction Network (GMCN), which constructs a scene-specific graph utilized by a Markov Random Field to optimize sparse depth through Gaussian Belief Propagation. Experimental results on the KITTI DC and NYU datasets demonstrate state-of-the-art performance, highlighting the effectiveness of the proposed approach.

**Strengths:**

1. The proposed method achieves state-of-the-art performance on both indoor and outdoor datasets.
2. It shows superior robustness across varying depth sparsity levels compared to existing approaches.
3. The paper provides a comprehensive analysis and extensive experimental results in the supplementary material, which further supports the validity of the proposed approach.

**Weaknesses:**

1. In Figure 1, it is recommended to add essential legends for better clarity, such as explaining the meaning of “T” in the top-middle and the significance of the green, blue, and orange lines.
2. The Method section currently occupies a substantial portion of the paper, leaving limited space for the Experiment section. It is suggested to compress the Method section to allow more room for presenting additional experimental results.
3. The influence of local edges and GBP iterations should be analyzed individually. Table 2 appears cluttered, making it difficult to identify corresponding variants. A clearer presentation or separate analysis would improve readability and understanding.

**Questions:**

1. In Table 1, could you clarify whether the entry labeled GBPN corresponds to the GBPN-1 or GBPN-2 variant?

---

> ### Author Response · Authors · 2025-11-21
> **Response to Reviewer p3v7**
>
> Thank you for recognizing our work and for your valuable feedback. We have carefully considered your comments and address your concerns below.
>
> **W1**: In Figure 1, it is recommended to add essential legends for better clarity.
>
> > **A**: Thanks for your suggestion. We have added legends in Figure 1 and introduced the meaning in the main text (line 150, line 211).
>
> >Specifically, "T" signifies the iteration count of the propagation process; the green and blue lines depict local and non-local edges (for serial and parallel propagation); the orange circles indicate depth measurements, and the gray circles signify the depth estimates.
>
>
>
> **W2**: Allow more room for presenting additional experimental results.
>
>
> > **A**: We have revised the manuscript to allocate more space to the experimental results. This includes streamlining the method section and adding new experiments as detailed below.
>
> >  **For revision**:, we compress the room for method part and extend the length of main paper from 9 pages to 10 pages. We rewrite the ablation studies section in a more clear way, and adding generalization analysis to the Experiment section (which is in the appendix previously). Correspondingly, we revise the abstract (line 26), introduction (line 54 \& 71), and conclusion section (line 538).
>
>
>
> **W3**: Table 2 appears cluttered, making it difficult to identify corresponding variants.
>
>
> >**A**: Thanks for your question. As suggested, we have significantly revised and clarified the Ablation Study section to demonstrate the contribution of each component more rigorously. In the revised version, we now present the ablation studies by introducing components incrementally, starting from a simple baseline ($V_1$) and progressively adding modules up to the final model ($V_9$). This approach ensures the configuration of each variant is explicit and the individual effectiveness of every component is clearly verifiable.
>
> > Specifically, we provide detailed configurations and analysis results starting from Line 112. We have also restructured Table 2 into four separate parts to facilitate a direct comparison of each core component's impact. For a comprehensive evaluation, we list both RMSE and the accuracy metric $\delta_{1.025}$. As shown in the updated results, moving from the baseline ($V_1$) to the final model ($V_9$), the RMSE drops from $342.70 mm$ to $100.69 mm$, and $\delta_{1.025}$ improves from $21.58\\%$ to $85.27\\%$. The large performance improvement demonstrates the strength and priority of our GBPN architecture and the effective contribution of each core component.
>
> > Furthermore, as detailed in Section A.2.2, we use Exponential Moving Average (EMA) during training, which contributes significantly to training stability. To verify the consistency of our model, we have retrained $V_9$ three times. The resulting RMSE values were $100.65 mm$, $100.69 mm$, and $100.74 mm$, demonstrating minimal variance and confirming the reliability of the results presented in Table 2. Therefore, these results accurately verify each component's contribution. Finally, as promised, we will release the code for training, testing and trained models once the paper is published.
>
> > **For revision**: To make the ablation study more clear, we have revised the main text with Table 2 in Section 4.2.
>
> **Q1**: In Table 1, could you clarify whether the entry labeled GBPN corresponds to the GBPN-1 or GBPN-2 variant?
>
> >**A**: Thanks for pointing this out. The entry labeled “GBPN” in Table 1 corresponds to the GBPN-2 variant. We have clarifies this in the revised version in line 377.
>
> Best Regards,
>
> Authors of submission 18865

---

> > ### Comment · Reviewer_p3v7 · 2025-11-24
> >
> > Thank the authors for their feedback. My concerns are addressed and I decide to keep my rating unchanged.

---

> > > ### Author Response · Authors · 2025-11-28
> > > **Response to Reviewer p3v7**
> > >
> > > Thank you for recognizing our work and for your valuable feedback.
> > > We are grateful for your thorough evaluation of our paper and for your active engagement during the discussion period.
> > > We are appreciated our responses have addressed all of your concerns.
> > > With the valuable suggestions from the rebuttal and discussions with all reviewers, our submission has been substantially enhanced.
> > > We would be delighted to respond to any further concerns you may have, to enhance our work sufficiently to warrant a higher score.
> > >
> > > Best Regards,
> > >
> > > Authors of submission 18865

---

### Official Review · Reviewer_684x · 2025-11-01

**Soundness:** 2
**Presentation:** 1
**Contribution:** 2
**Rating:** 4
**Confidence:** 3

**Summary:**

This paper introduces a hybrid framework, termed the Gaussian Belief Propagation Network (GBPN), for depth completion using sparse depth points and color images. The core idea is to use a deep network (GMCN) to dynamically construct a Markov Random Field (MRF) for each scene, learning both its potential function and graph structure by predicting non-local edges. Subsequently, the Gaussian Belief Propagation (GBP) algorithm is employed to infer dense depth from the constructed MRF. The authors report that the proposed method achieves state-of-the-art performance on the NYUv2 and KITTI datasets and exhibits strong robustness to input sparsity.

**Strengths:**

1.	Framing deep completion as probabilistic inference on dynamically constructed graph models offers a theoretically sound approach for handling sparse and irregular inputs. Extensive evaluations under varying sparsity levels, noise conditions, and cross-dataset settings show that the proposed framework achieves stronger robustness than pure end-to-end regression models.
2.	The proposed method not only learns the MRF parameters but also infers the graph structure by predicting non-local edges, which represents a novel contribution. This design allows the model to adaptively capture long-range dependencies from image content, thereby overcoming the fixed-neighborhood constraints of traditional MRFs.

**Weaknesses:**

1.	The ablation study (Table 2) is presented in not clear, making it difficult to verify the contribution of each model component.
2.	Although the paper provides a thorough empirical comparison with competitors such as BP-Net and demonstrates clear advantages in accuracy and robustness, the discussion does not move beyond empirical evidence and lacks a compelling conceptual justification. The authors do not clearly explain why their MRF+GBP paradigm is theoretically or conceptually superior to the direct learning propagation paradigm represented by BP-Net. The contributions appear to represent a highly successful and well-designed paradigm instantiation rather than a fundamental conceptual advancement.
3.	The inference time of this method is considerably longer than that of its main competitors (nearly 80% slower than BP-Net on KITTI), yet the accuracy improvement is negligible (only about 0.4%). This trade-off is unacceptable for real-time applications such as autonomous driving.
4.	This paper employs loopy belief propagation on dynamically generated graphs, a method that lacks formal convergence guarantees. However, the paper does not discuss or analyze the potential stability and convergence issues associated with this setting.

**Questions:**

1. Could you provide a clearer version of Table 2 that explicitly lists the configuration details for each ablation study variant?
2. For practical applications such as autonomous driving, how do you justify a considerable increase in inference latency in exchange for only a marginal gain in accuracy?
3. What conceptual advantages does your approach offer over methods that directly learn propagation operators?
4. When applying loopy belief propagation to dynamically generated graphs, have you observed any cases of non-convergence or oscillation?

---

> ### Author Response · Authors · 2025-11-21
> **Response to Reviewer 684x [Part 1/4]**
>
> Thank you for your valuable feedback. We have carefully considered your comments and address your concerns below.
>
> **W1 \& Q1**: Could you provide a clearer version of the ablation study?
>
>
> >**A**: Thanks for your question. As suggested, we have significantly revised and clarified the Ablation Study section to demonstrate the contribution of each component more rigorously. In the revised version, we now present the ablation studies by introducing components incrementally, starting from a simple baseline ($V_1$) and progressively adding modules up to the final model ($V_9$). This approach ensures the configuration of each variant is explicit and the individual effectiveness of every component is clearly verifiable.
>
> > Specifically, we provide detailed configurations and analysis results starting from Line 411. We have also restructured Table 2 into four separate parts to facilitate a direct comparison of each core component's impact. For a comprehensive evaluation, we list both RMSE and the accuracy metric $\delta_{1.025}$. As shown in the updated results, moving from the baseline ($V_1$) to the final model ($V_9$), the RMSE drops from $342.70 mm$ to $100.69 mm$, and $\delta_{1.025}$ improves from $21.58 \\%$ to $85.27 \\%$. The large performance improvement demonstrates the strength and priority of our GBPN architecture and the effective contribution of each core component.
>
> > Furthermore, as detailed in Section A.2.2, we use Exponential Moving Average (EMA) during training, which contributes significantly to training stability. To verify the consistency of our model, we have retrained $V_9$ for three times. The resulting RMSE values were $100.65 mm$, $100.69 mm$, and $100.74 mm$, demonstrating minimal variance and confirming the reliability of the results presented in Table 2. Therefore, these results accurately verify each component's contribution. Finally, as promised, we will release the code for training, testing and trained models once the paper is published.
>
> > **For revision**: To make the ablation study more clear, we have revised the main text with Table 2 in Section 4.2.

---

> ### Author Response · Authors · 2025-11-21
> **Response to Reviewer 684x [Part 2/4]**
>
> **W2 \& Q3**: What conceptual advantages does your approach offer over methods that directly learn propagation operators?
>
> >**A**: We thank the reviewer for raising this important point regarding the conceptual distinctions between our GBPN and other propagation-based methods. We appreciate the opportunity to provide a detailed explanation. We will first elaborate on the key differences **in detail** and then provide a **summary**.
>
> >**Explaining in detail**: Firstly, existing propagation-based methods still face challenges in processing sparse data. Originally introduced by CSPN, propagation-based methods for depth completion, including their variants like CSPN++, DyPN, and NLSPN, primarily function as post-processing refinement modules. Consequently, they require an initial depth estimate from a deep network, which itself faces the challenge in processing sparse data. BP-Net attempts to alleviate this by propagating depth in a pre-processing stage. However, its use of a small MLP for propagation yields insufficiently accurate results, necessitating additional modules like Multi-modal Fusion Network and CSPN-based post-processing for further refinement. In contrast, our GBPN introduces a unified framework that synergistically integrates deep learning with probabilistic graphical models. Sparse depth measurements are naturally incorporated as principled data terms within a globally consistent Markov Random Field (MRF), inherently addressing input sparsity and irregularity without requiring auxiliary modules.
>
>
> > Secondly, current propagation-based methods suffer from a limited information propagation range. Whether using local convolutions (as in CSPN variants) or nearest-neighbor MLPs (as in BP-Net), the propagation process is confined to local regions. Sparse depth information cannot reliably propagate across the entire image. For instance, in BP-Net, a pixel can only receive information from its four nearest depth measurements. In contrast, our GBPN guarantees that the depth at any pixel can influence, and be influenced by, information from the entire image space in just one iteration. The relationship between GBPN and other propagation-based methods is analogous to that of Transformers and CNNs: just as Transformers overcome the limited receptive field of CNNs in feature extraction, GBPN overcomes the limited propagation range in depth completion.
>
>
> >Thirdly, while typical propagation-based methods yield only a dense depth map (a point estimate in statistics), our GBPN estimates a full depth distribution, providing both a depth value and a confidence measure for each pixel. As introduced in line 058, the confidence map is highly valuable for downstream, risk-aware applications such as robotic planning.
>
>
> >**In summary**: Other propagation-based methods are limited by local propagation ranges and typically operate in a single stage (either pre- or post-processing), requiring combination with other modules to complete the task. In contrast, our GBPN is a unified framework for depth distribution estimation. It achieves global information propagation, allowing depth from any pixel to inform the entire image, without relying on other components.
>
>
> >**For Revision**:  we have revised the introduction section(line 054) to highlight the advantages. And emphasis the issue we solved compared to previous methods in related work section (line 124).

---

> ### Author Response · Authors · 2025-11-21
> **Response to Reviewer 684x [Part 3/4]**
>
> **W3 \& Q2**: How do you justify a considerable increase in inference latency in exchange for only a marginal gain in accuracy?
>
>
> > **A**: Thank you for this question. We wish to highlight that GBPN improves not only in accuracy but also significantly in robustness and generalization capability, which are as critical as accuracy, if not more, for real-world applications. Furthermore, the observed inference latency is not primarily due to increased computational demands, but rather stems from our current naive implementation. We acknowledge this issue and plan to address it through optimization implementation in the future work.
>
> > **For performance**: GBPN delivers consistently strong results across both the KITTI and NYUv2 benchmarks. It outperforms the strong competitor BP-Net on all evaluation metrics (RMSE, MAE, REL, $\delta_{1.02}$, etc.). Notably, GBPN achieves substantial improvement on the NYUv2 dataset, obtaining the best RMSE and $\delta_{1.02}$. As shown in Table 1, compared to BP-Net, the $\delta_{1.02}$ of GBPN improves from $87.2\\%$ to $89.1\\%$ (a $1.9\\%$ absolute gain). Tables 6 and 7 further demonstrate that GBPN consistently surpasses all competing methods, achieving the lowest errors under each LiDAR line or point density setting. This advantage is particularly pronounced in extremely sparse regimes (e.g., 8-line KITTI and 20–100 points on NYUv2). For instance, in Table 6, with 8-line LiDAR on KITTI, compared to BP-Net, GBPN improves the RMSE from $4541.9 mm$ to $2750.4 mm$ (an absolute improvement of $1791.5 mm$). Similarly, in Table 7, with 20 points on NYUv2, compared to BP-Net, GBPN improves the RMSE from $0.749 m$ to $0.649 m$ (a $0.1 m$ absolute gain). Moreover, in a zero-shot cross-dataset evaluation on the VOID benchmark, GBPN achieves the best RMSE and MAE across all sparsity levels, demonstrating superior generalization capability across varying sparsity level, sparsity patterns, and scenes. As listed in Table 3, with 1500 sparse points on VOID, compared to BP-Net, GBPN improves the RMSE from $0.74 m$ to $0.68 m$ (a $0.06 m$ absolute gain). In summary, considering the comprehensive gains in accuracy, robustness, and generalization, the performance improvement of GBPN is substantial and not marginal compared to any competitor.
>
> > **For latency**: As shown in Table 9, our method exhibits moderate latency among current depth completion methods. For instance, CFormer and OGNI both require more inference time than GBPN. We have also analyzed the computation, parameters, and runtime of GBPN in Section A.7. Our approach adopts a small, lightweight network as its backbone. However, as shown in Table 10, the optimization stage, while computationally less intensive than the deep network, occupies most of the runtime. This efficiency bottleneck is primarily due to our custom, naive implementation of Gaussian Belief Propagation (GBP). Compared to highly-optimized official PyTorch layers, our GBP implementation has significant room for improvement, and we are committed to optimizing it in future work.
>
>
> > **For Revision**: We have revised the Abstract section (line 026), Introduction section (line 071) to highlight the substantial performance improvement of GBPN. We change the section about generalizable capability from appendix to the main paper(section 4.4) to make the main paper self-contain. We add more analysis on the computation and latency in section A.7 (line 1390) and Table 11. We also promise to optimize the implementation in Conclusion section (line 538).

---

> ### Author Response · Authors · 2025-11-21
> **Response to Reviewer 684x [Part 4/4]**
>
> **W4 \& Q4**: The potential stability and convergence issue on loopy belief propagation.
>
> > **A**: Thank you for this important question. We have made two specific design choices in our GBPN for keeping stability and convergence. First, as detailed in Section 3.3.1, while the overall MRF contains loops, we decompose it into several loop-free sub-graphs for each time gaussian belief propagation. This decomposition inherently mitigates the influence of loops. Furthermore, we employ the damping technique during message passing, which is a well-established practice to enhance stability and convergence[1]. And, we did not observe any instability or non-convergence issues when applying GBP to our dynamically generated graphs.
>
>
> > Besides, while it is true that LBP lacks formal convergence guarantees for general graphs, it has been empirically validated as highly reliable in numerous practical applications[3]. Even in loopy graphs, LBP often converges to a good approximation of the optimal solution. As noted in the literature[2], even when it does not reach the global optimum, it typically converges to a useful sub-optimal solution.
>
> > Moreover, we note that this situation is not unique to LBP. Many successful optimization algorithms, such as the Alternating Direction Method of Multipliers (ADMM), are widely used despite similar theoretical limitations. They are employed because they demonstrate robust and effective convergence in practice. We view the use of LBP in our framework in a similar light—as a powerful and empirically reliable tool for our specific task.
>
>
> >  **For Revision**: To make the potential stability and convergence issue more clear, we discuss and analyze the correspond issues in the Method section (line 224, line 255, line 266, line 273).
>
>
>
> **Ref:**
> >[1] Kevin Murphy et al., Loopy belief propagation for approximate inference: An empirical study.
>
> >[2] Bickson D et al., Gaussian belief propagation: Theory and aplication.
>
> >[3] Yedidia J S et al., Constructing free-energy approximations and generalized belief propagation algorithms.
>
>
> Best Regards,
>
> Authors of submission 18865

---

> ### Author Response · Authors · 2025-11-28
> **Kindly Remind to Reviewer 684x**
>
> Thank you sincerely for your thoughtful and constructive feedback. As the discussion period is drawing to a close,
> we'd like to kindly remind we have addressed all your concerns in the response.
> Also, we are delighted to respond to any further concerns you may have,
>  to enhance our work sufficiently to warrant a higher score.
>
> Best Regards,
>
> Authors of submission 18865

---

### Author Response · Authors · 2025-12-01
**Final Remarks by Authors**

The **theory soundness**, **SOTA performance**, **strong robustness** of our method are commonly acknowledged by all the reviewers. All concerns have been addressed in detail and the manuscript correspondingly revised, satisfying the active reviewers. There's **no critical attack** remaining against this submission. We express our heartfelt thanks to all reviewers for their valuable time, and regret that the discussion was unexpectedly interrupted.

In summary, our method presents a **novel hybrid framework** that synergistically integrates deep learning with probabilistic graphical models for end-to-end depth completion (DC). Our approach successfully overcomes the challenges in current DC methods related to **sparse data processing** and **limited information propagation range**. Consequently, our method demonstrates superior performance, notable robustness, and strong generalizable capability.

We sincerely believe this work will be of high interest to the community.
Thanks for your consideration.

Best regards,

Authors of submission 18865

---

### Meta-Review · Area_Chair_dVvW · 2026-01-12

**Summary:**

Overall, reviewers agreed that the paper is technically sound and well executed, but does not yet reach the level expected for acceptance at a major conference. The main concerns are that the empirical improvements over existing strong baselines are relatively modest and sometimes inconsistent, especially when considering the additional complexity and computational cost introduced by the method. Several reviewers questioned whether the performance gains sufficiently justify the increased inference time and system complexity, particularly for practical or real-time applications. There were also concerns about the maturity and positioning of the work, including whether the current version fully reflects and integrates feedback from prior review cycles. While many clarity and presentation issues were addressed in the rebuttal, these remaining concerns led reviewers to view the paper as solid but not yet compelling enough for acceptance at a top-tier venue.

**Reviewer Concerns:**

The rebuttal addressed most concerns related to clarity, presentation, and empirical validation. In particular, the authors clarified the ablation study, improved figure and table readability, explained the serial and parallel propagation scheme, added analysis on robustness and cross-dataset generalization, and provided a clearer conceptual comparison with propagation-based methods such as BP-Net and CSPN. These responses resolved the main concerns of Reviewers p3v7 and NRg4, and partially addressed rfq1’s concerns about ablation clarity and sparsity behavior.

However, some concerns remain. The main outstanding issue is that the ICLR submission did not sufficiently reflect feedback from the prior NeurIPS review at the time of submission, which Reviewer rfq1 continued to view as a significant problem. In addition, concerns about limited conceptual novelty, small empirical gains relative to model complexity, and the lack of a strong justification for the efficiency–accuracy trade-off were not fully resolved. Formal convergence guarantees for loopy belief propagation also remain an open theoretical issue.

**Reviewer Scores:**

Reviewer p3v7 and Reviewer NRg4 explicitly stated after the rebuttal and discussion that they would keep their original scores of 6 unchanged, as they felt their concerns had been adequately addressed but not in a way that warranted a higher score. Reviewer rfq1 initially gave a score of 2 but raised it to 4 after the authors’ detailed responses and revisions. However, this reviewer still expressed a major concern that the ICLR submission did not sufficiently reflect the comments from the prior NeurIPS review, which appears to have prevented a further increase. Reviewer 684x did not explicitly update their score during the discussion, and given their remaining concerns regarding conceptual novelty, runtime trade-offs, and the stability of loopy belief propagation, it is likely that their score would have remained around its original level (approximately 4).

---

### Decision · Program_Chairs · 2026-01-26

Reject